# Versatile roles of rubredoxin reductase of *Pseudomonas aeruginosa* TBCF10839 in virulence and stress protection

Lutz Wiehlmann[1,2,3], Thorsten Adams[1¤a], Claus Urbanke[2], Sonja Horatzek[1¤b], Doris Jordan[4¤c], Ilona Rosenboom[1,3], Leo Eberl[5], Burkhard Tümmler[1,2,6]*

1 Department of Pediatric Pneumology, Allergology and Neonatology, Hannover Medical School, Hannover, Germany, 2 Department of Biophysical Chemistry, Hannover Medical School, Hannover, Germany, 3 Research Core Unit Genomics, Hannover Medical School, Hannover, Germany, 4 Institute for Medical Microbiology and Hospital Epidemiology, Hannover Medical School, Hannover, Germany, 5 Department of Plant and Microbial Biology, University of Zurich, Zurich, Switzerland, 6 Biomedical Research in Endstage and Obstructive Lung Disease, German Center for Lung Research, Hannover, Germany

¤a Current address: Sartorius Stedim Biotech GmbH, Göttingen, Germany.
¤b Current address: VW Kraftwerk GmbH, Wolfsburg, Germany.
¤c Current address: Kleintierpraxis Dr. Doris Jordan, Melle, Germany.
* tuemmler.burkhard@mh-hannover.de

## Abstract

Many microorganisms can degrade alkanes, using them as carbon source. The first and key step in alkane utilization is its hydroxylation, which requires a catalytic membrane-bound monooxygenase, a soluble rubredoxin and a soluble rubredoxin reductase. By comparing the phenotype of *Pseudomonas aeruginosa* strain TBCF10839 with an isogenic mutant that carries a plasposon within the rubredoxin reductase gene *rubB* (PA5349) and the complemented mutant, we report multiple, yet unknown roles of rubredoxin reductase in the physiology of *P. aeruginosa* beyond alkane hydroxylation. The plasposon mutant TBCF10839 *rubB*::Tn*5* was severely compromised in its versatility to degrade protein and did not produce any *N*-acyl homoserine lactone signal molecules. Consequently, the quorum-sensing deficient mutant was avirulent in the *Caenorhabditis elegans* fast killing infection model. An intact *rubB* gene was essential for the TBCF10839 strain to inactivate hydrogen peroxide and to persist and multiply in human neutrophils. Upon exposure to hydrogen peroxide, the ternary complex of rubredoxin reductase, rubredoxin and catalase initially mediates the direct reduction to water followed by disproportionation into water and oxygen when the NADH pool is depleted. In summary, *P. aeruginosa* TBCF10839 engages the electron-transfer proteins rubredoxin reductase and rubredoxin for stress protection and virulence.

**Data availability statement:** All data are in the manuscript and the supporting information files. The sequence of the genome of the P. aeruginosa strain TBCF10839 can be found at the NCBI database, accession number PRJNA975170. More microarray hybridization data about the gene expression of wild type strain TBCF10839 are available at the GEO database (accession no. GSE36647).

**Funding:** This work was supported by the Deutsche Forschungsgemeinschaft (https://www.dfg.de/) (SFB 587, A9) to BT. DJ was a member of the DFG-sponsored International German-Danish Research Training Group 'Pseudomonas: Pathogenicity and Biotechnology' (IRTG 653). SH received a predoctoral Lichtenberg stipend from the Niedersächsisches Ministerium für Wissenschaft und Kultur. The funders had no role in study design, data collection and analysis, decision to publish, or preparation of the manuscript.

**Competing interests:** The authors declare that the research was conducted in the absence of any commercial or financial relationships that could be construed as a potential conflict of interest. Thus, the authors have declared that no competing interests exist.

## Authors summary

Rubredoxin and rubredoxin reductase are phylogenetically ancient iron sulfur proteins that protect anaerobic bacteria from the deleterious effect of oxygen. Moreover, many bacteria engage this electron transfer and redox system for the degradation of alkanes as carbon source. Here we report on multiple further, yet unknown roles of rubredoxin and rubredoxin reductase in metabolism, signaling, secretion and virulence of the opportunistic pathogen *Pseudomonas aeruginosa*. Rubredoxin reductase controls the oxidation state of iron sulfur proteins in the *P. aeruginosa* cell, confers resistance against oxidative stress and facilitates bacterial growth and persistence in neutrophils, man's major antipseudomonal weapon. In other words, rubredoxin and rubredoxin reductase are a showcase of how key elements of cellular protection and metabolism may become bacterial virulence determinants that undermine human defense in infectious disease.

## Introduction

*Pseudomonas aeruginosa* is an environmental microorganism that preferentially thrives in inanimate aquatic habitats at low frequency [1] and can colonize the animate surfaces of plants, animals and humans [2]. The inanimate habitats harbor the largest pool of clones out of which subgroups will spread to more specialized niches such as the mucosal surfaces of the predisposed human host [3,4]. Thanks to its global distribution, humans are regularly exposed to a few *P. aeruginosa* bacteria making it a transient member of the oral and airway microbiome at very low abundance [5]. *P. aeruginosa* is classified as an opportunistic pathogen equipped with a large repertoire of virulence determinants [2] and a complex regulatory network of intracellular and intercellular signals [6] that allow the bacteria to adapt to an animate niche and to escape host defense.

The metabolically versatile organism outcompetes other microbes in nutrient-poor habitats such as the hospital environment [7–9]. For example, *P. aeruginosa* can utilize alkanes, soaps, detergents and disinfectants as carbon source [10–13].

Alkanes are chemically inert apolar molecules. The first and key step in alkane utilization is its hydroxylation [14,15], which requires an electron transfer and redox system, consisting of a catalytic membrane-bound monooxygenase, a soluble rubredoxin and a soluble rubredoxin reductase [16–18]. Protein structures have been resolved for the alkane monooxygenase of *Fontimonas thermophila* [19,20] and the rubredoxin – rubredoxin reductase complex of *Pseudomonas aeruginosa* [21], providing a reaction model for the oxidation of the terminal carbons.

Alkanes diffuse into the bacterial membrane and are incorporated into a hydrophobic substrate binding channel terminating at the catalytic non-heme diiron center of the monooxygenase [22]. Electrons for the reaction are provided from the donor NADH through an electron transport chain and are transferred within rubredoxin reductase from the NAD(P)H binding cavity via the FAD binding pocket to the rubredoxin binding site [21]. Rubredoxin is reduced and transfers electrons to the iron

center of the monooxygenase promoting binding and activation of $O_2$ and formation of the terminal hydroxyl group via a Fe-O intermediate [19,20,23].

The *P. aeruginosa* genome harbors two alkane-1-monooxygenase genes, *alkB1* (PA2574) and *alkB2* (PA1525), and a contiguous cluster of one rubredoxin reductase, *rubB* (PA5349), and two rubredoxin genes, *rubA2* (PA5350) and *rubA1* (PA5351). Consistent with the different organization and chromosomal locations of the genes, RubA1, RubA2 and RubB do not match in their expression and substrate profiles with those of the alkane hydroxylases. AlkB1 and AlkB2 are long-chain C(10) to C(24) alkane hydroxylases, whereas RubA1, RubA2 and RubB are more promiscuous and also promote the oxidation of medium-chain alkanes such as n-octane [24]. The expression of AlkB1 and AlkB2 is alkane-dependent, but RubB/RubA1/RubA2 are constitutively expressed in absence and presence of alkanes [10].

The alkane-independent mode of gene expression indicates that the complex of rubredoxin reductase and rubredoxin fulfills a more general role of electron transfer in *P. aeruginosa* besides its essential function as an electron shuttle in alkane hydroxylation. We have constructed a non-auxotrophic plasposon library of the clinically virulent *P. aeruginosa* strain TBCF10839 [25] to search for genes that are essential for habitat-specific bacterial survival. By screening the library for mutants that are more susceptible to killing by human neutrophils in phagocytosis assays, we identified a strongly attenuated mutant that carried the plasposon within the rubredoxin reductase gene *rubB* (PA5349). By comparing wild type strain with the plasposon mutant and the complemented mutant, we unraveled yet unknown multiple roles of rubredoxin reductase in the physiology of *P. aeruginosa* beyond alkane hydroxylation.

## Results

### *P. aeruginosa* **TBCF10839 is proficient to grow on alkanes**

*P. aeruginosa* TBCF10839 is a clinical isolate from the airways of an individual with cystic fibrosis [26]. Thus, we first tested whether the clinical isolate TBCF10839 like any environmental isolate can use alkanes as carbon source. As shown in Fig 1A, strain TBCF10839 grows on hexadecane as sole carbon source. The mutant TBCF10839 *rubB*::Tn*5* that carries a transposon insertion in the 1,155 bp large rubredoxin reductase gene at position 977 is unable to utilize hexadecane. Complementation of the mutant with the plasmid pME6010::TB*rubB in trans* restored the ability to grow on hexadecane demonstrating that the rubredoxin reductase RubB is necessary for alkane degradation by the TBCF10839 strain.

Cell pellets of bacteria grown in LB until the onset of quorum sensing (QS) during stationary phase were either 'linen tan' (wt TBCF10839), 'dusky pink' (TBCF10839 *rubB*::Tn*5*) or 'sand tan' (complemented TBCF10839 *rubB*::Tn*5*; pME6010::TB*rubB*), indicating that rubredoxin reductase has a global role for TBCF10839 to control the oxidation state of iron centers in the cell (Fig 1B). Consistent with this interpretation, the cell pellet of mutant TBCF10839 *rubB*::Tn*5* became 'linen tan' like wild type if the cell suspension was incubated with the reducing agent ascorbic acid (Fig 1C).

### Gene chip comparison of TBCF10839 and the TBCF10839 *rubB*::Tn*5* mutant

To investigate whether the rubredoxin reductase/ rubredoxin system has a function beyond alkane degradation, we compared the global expression of the core genome of the TBCF10839 *rubB*::Tn*5* mutant and its TBCF10839 parent. The cultures were grown aerobically side-by-side in ABC minimal medium (two biological replicates in triplicate). Total RNA was extracted and hybridized onto *P. aeruginosa* PAO1 Gene Chips (complete dataset in Supporting Information, S1 Table), which cover 87.7% of the 6.9 Mbp large TBCF10839 genome with 99.63% nucleotide sequence identity [26–28] (For previously published Gene Chip comparisons of gene expression profiles of wild type TBCF10839 with isogenic transposon mutants, please see references [26,27,29–32]).

Compared to its wild type parent, the TBCF10839 *rubB*::Tn*5* mutant significantly upregulated the mRNA transcript expression of 67 genes of the core genome by more than 7-fold (Table 1). Fifty-nine of these 67 genes were only weakly or marginally expressed in wild type TBCF10839. Besides numerous hypotheticals of unknown function, the mutant

A

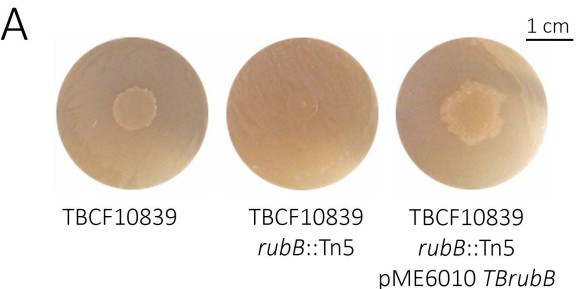

1 cm

TBCF10839 TBCF10839 TBCF10839
*rubB*::Tn5 *rubB*::Tn5
pME6010 *TBrubB*

B

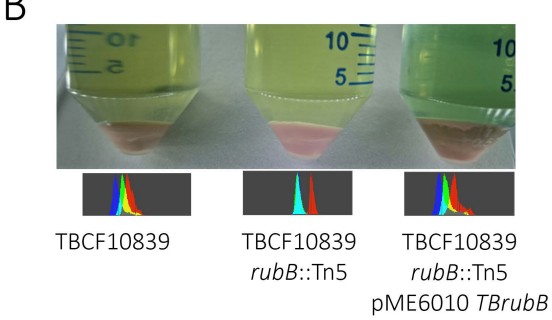

TBCF10839 TBCF10839 TBCF10839
*rubB*::Tn5 *rubB*::Tn5
pME6010 *TBrubB*

C

TBCF10839 *rubB*::Tn5

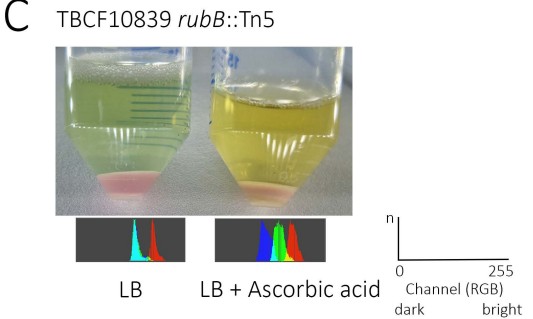

n

LB LB + Ascorbic acid 0 255
 Channel (RGB)
 dark bright

**Fig 1. A. Plate assay. Growth of *P. aeruginosa* exposed to vapor n-hexadecane as sole carbon source. In contrast to the isogenic transposon mutant TBCF10839 *rubB*::Tn5 (middle), wild type TBCF10839 (left) and the complemented transposon mutant TBCF10839 *rubB*::Tn5; pME6010::TB*rubB* (right) are able to utilize hexadecane as the sole carbon source (n=10 biological replicates). B.** Cell pellets (from left to right) of TBCF10839, TBCF10839 *rubB*::Tn5 and TBCF10839 *rubB*::Tn5; pME6010::TBrubB grown to the onset of quorum sensing during stationary phase (n > 100 biological replicates, phenotype always observed whenever bacterial cultures were pelleted). **C.** Cell pellets of mutant TBCF10839 *rubB*::Tn5 in the absence (left) and presence (right) of 30 mM ascorbic acid (n=5 biological replicates). **B, C.** The color of the cell pellets is visualized by photographs (upper panel) and windows of 100x200 pixels of RGB channels taken from the middle of the pellet (lower panel). The coordinate system on the right indicates the meaning of the axes: y-axis: n, number of dots of the respective RGB channels; x-axis: brightness values of the red, green and blue channels ranging (from left to right) from 0 to 255.

upregulated the expression of transcripts required for the utilization of lipids. The maximal increase of transcript levels was observed for the ribosome modulation factor Rmf that upon the induction by the alarmone ppGpp orchestrates the cellular stress response and creates ribosome dimers in stationary phase [33].

annotation according to the entries in the Pseudomonas Genome Database, version 22.1, accessed May 25th, 2025.

Whereas no function could be assigned to the majority of genes that were upregulated in the mutant, 85% of the 165 genes were annotated that were downregulated in TBCF10839 *rubB*::Tn5 by more than seven-fold (Table 1). The gene list comprises key members of energy metabolism, intermediary metabolism and the translational apparatus such as the ATP synthase, succinate dehydrogenase and ribosomal proteins suggesting that TBCF10839 *rubB*::Tn5 reduces the

**Table 1. Gene chip comparison of TBCF10839 and the TBCF10839 *rubB*::Tn*5* mutant grown aerobically in ABC medium to early stationary phase.**

| | | Descriptions | TBCF10839 *rubB*::Tn*5* |
|---|---|---|---|
| | | | upregulated |
| PA0022 | | conserved hypothetical protein | 10.2 |
| PA0109 | | hypothetical protein | 11.5 |
| PA0131 | *BauB* | Cupin-2 domain, part of bauDCBA operon | 19.6 |
| PA0132 | *BauA* | beta-alanine--pyruvate transaminase | 7.8 |
| PA0323 | | probable binding protein component of ABC transporter | 12.7 |
| PA0441 | *dht* | dihydropyrimidinase | 13.4 |
| PA0476 | | probable permease | 13.6 |
| PA0613 | | hypothetical protein | 24.7 |
| PA0842 | | probable glycosyl transferase | 9.7 |
| PA0850 | | hypothetical protein | 17.8 |
| PA0865 | *hpd* | 4-hydroxyphenylpyruvate dioxygenase | 25.4 |
| PA0911 | *alpE* | AlpE, part of AlpABCDE operon, response to DNA damage | 8.6 |
| PA0921 | | hypothetical protein | 7.7 |
| PA0922 | | hypothetical protein | 8.9 |
| PA1014 | *wapB* | 1,2-glucosyltransferase WapB | 8.9 |
| PA1077 | *flgB* | flagellar basal-body rod protein FlgB | 9.5 |
| PA1136 | | probable transcriptional regulator | 10.6 |
| PA1225 | | FAD-dependent NADPH:quinone reductase | 10.4 |
| PA1264 | | probable transcriptional regulator | 12.2 |
| PA1302 | *hxuA* | heme receptor HxuA | 13.7 |
| PA1420 | | hypothetical protein | 10.3 |
| PA1438 | | probable two-component sensor | 8.1 |
| PA1500 | | probable oxidoreductase | 17.1 |
| PA1514 | | ureidoglycolate hydrolase YbbT | 13.0 |
| PA1761 | | hypothetical protein | 13.5 |
| PA1762 | | hypothetical protein | 17.9 |
| PA1942 | | hypothetical protein | 9.3 |
| PA1984 | *exaC* | NAD+ dependent aldehyde dehydrogenase ExaC | 10.9 |
| PA2035 | | probable decarboxylase | 7.8 |
| PA2046 | | hypothetical protein | 9.3 |
| PA2074 | | hypothetical protein | 8.6 |
| PA2100 | | probable transcriptional regulator | 13.1 |
| PA2219 | *opdE* | membrane protein OpdE | 10.8 |
| PA2221 | | conserved hypothetical protein | 8.9 |
| PA2457 | | hypothetical protein | 12.2 |
| PA2481 | | hypothetical protein | 12.8 |
| PA2862 | *lipA* | lactonizing lipase precursor | 14.0 |
| PA2920 | | probable chemotaxis transducer | 13.0 |
| PA3049 | *rmf* | ribosome modulation factor | 15.5 |
| PA3080 | | hypothetical protein | 10.3 |
| PA3145 | *wbpL* | glycosyltransferase WbpL | 22.2 |
| PA3181 | | 2-keto-3-deoxy-6-phosphogluconate aldolase | 18.8 |
| PA3252 | | probable permease of ABC transporter | 10.3 |
| PA3369 | | hypothetical protein | 9.1 |

*(Continued)*

**Table 1.** (Continued)

| | | Descriptions | TBCF10839 *rubB*::Tn5 |
|---|---|---|---|
| PA3370 | | hypothetical protein | 14.4 |
| PA3416 | | probable pyruvate dehydrogenase E1 component, beta chain | 38.7 |
| PA3417 | | probable pyruvate dehydrogenase E1 component, alpha subunit | 13.4 |
| PA3509 | | probable hydrolase | 9.0 |
| PA3581 | glpF | glycerol uptake facilitator protein | 10.1 |
| PA3584 | *glpD* | glycerol-3-phosphate dehydrogenase | 13.7 |
| PA3919 | | conserved hypothetical protein | 62.1 |
| PA3972 | | probable acyl-CoA dehydrogenase | 8.4 |
| PA4023 | *eat* | ethanolamine transporter, Eat | 27.9 |
| PA4182 | | hypothetical protein | 7.7 |
| PA4326 | | hypothetical protein | 9.7 |
| PA4690 | | frameshift hypothetical protein | 8.7 |
| PA4813 | *lipC* | lipase LipC | 10.3 |
| PA5091 | *hutG* | N-formylglutamate amidohydrolase | 7.9 |
| PA5303 | | conserved hypothetical protein | 11.7 |
| PA5313 | *gabT2* | transaminase | 9.5 |
| PA5325 | *sphA* | SphA, induced by sphingosine | 10.3 |
| PA5352 | | conserved hypothetical protein | 9.9 |
| PA5380 | *gbdR* | GbdR, regulator of choline metabolism | 8.9 |
| PA5444 | | conserved hypothetical protein | 9.9 |
| | | Descriptions | |
| | | | downregulated |
| PA0195 | *pntA* | pyridine nucleotide transhydrogenase alpha subunit | -8.9 |
| PA0263 | *hcpC* | secreted protein Hcp | -25.2 |
| PA0547 | | probable transcriptional regulator | -12.5 |
| PA0548 | *tktA* | Transketolase | -13.9 |
| PA0595 | *lptD* | LPS-assembly protein LtpD | -14.3 |
| PA0649 | *trpG* | anthranilate synthase component II | -9.0 |
| PA0837 | *slyD* | peptidyl-prolyl cis-trans isomerase SlyD | -12.0 |
| PA0852 | *cpbD* | chitin-binding protein CbpD precursor | -21.0 |
| PA0903 | *alaS* | alanyl-tRNA synthetase | -26.5 |
| PA0943 | | hypothetical protein | -10.8 |
| PA0996 | *pqsA* | PqsA | -14.9 |
| PA0997 | *pqsB* | PqsB | -14.0 |
| PA0998 | *pqsC* | PqsC | -11.4 |
| PA1000 | *pqsE* | PqsE, Quinolone signal response protein | -38.6 |
| PA1002 | *phnB* | anthranilate synthase component II | -17.8 |
| PA1011 | | hypothetical protein | -11.9 |
| PA1045 | | hypothetical protein | -8.2 |
| PA1076 | | hypothetical protein | -10.2 |
| PA1168 | | hypothetical protein | -26.1 |
| PA1169 | | probable lipoxygenase | -40.5 |
| PA1278 | *cobP* | cobinamide kinase | -9.9 |
| PA1431 | *rsaL* | regulatory protein RsaL | -62.6 |
| PA1432 | *lasI* | autoinducer synthesis protein LasI | -10.2 |

*(Continued)*

| | | Descriptions | TBCF10839 *rubB*::Tn5 |
|---|---|---|---|
| PA1493 | *cysP* | sulfate-binding protein of ABC transporter | -12.7 |
| PA1546 | *hemN* | oxygen-independent coproporphyrinogen III oxidase | -49.6 |
| PA1552 | *ccoP1* | cytochrome c oxidase, cbb3-type, CcoP subtype | -26.2 |
| PA1556 | *ccoO2* | cytochrome c oxidase, cbb3 type, CcoO subtype | -24.1 |
| PA1581 | *sdhC* | succinate dehydrogenase (C subunit) | -13.0 |
| PA1582 | *sdhD* | succinate dehydrogenase (D subunit) | -9.9 |
| PA1583 | *sdhA* | succinate dehydrogenase (A subunit) | -26.0 |
| PA1588 | *sucC* | succinyl-CoA synthetase beta chain | -10.6 |
| PA1598 | | conserved hypothetical protein | -9.7 |
| PA1657 | *hsiB2* | HsiB2 | -61.9 |
| PA1658 | *hsiC2* | HsiC2 | -37.2 |
| PA1665 | *fha2* | Fha2 | -13.6 |
| PA1667 | *hsiJ2* | HsiJ2 | -19.0 |
| PA1669 | *icmF2* | IcmF2 | -13.5 |
| PA1757 | *thrH* | homoserine kinase | -10.7 |
| PA1789 | | hypothetical protein | -12.8 |
| PA1800 | *tig* | trigger factor | -29.9 |
| PA1812 | *mltD* | membrane-bound lytic murein transglycosylase D precursor | -19.0 |
| PA1837 | | hypothetical protein | -9.3 |
| PA1869 | *acp1* | Acp1, acyl carrier protein | -11.8 |
| PA1873 | | hypothetical protein | -9.4 |
| PA1901 | *phzC* | phenazine biosynthesis protein PhzC | -47.7 |
| PA1902 | *phzD* | phenazine biosynthesis protein PhzD | -21.1 |
| PA1903 | *phzE* | phenazine biosynthesis protein PhzE | -35.7 |
| PA1904 | *phzF2* | probable phenazine biosynthesis protein | -8.5 |
| PA1905 | *phzG2* | probable pyridoxamine 5 -phosphate oxidase | -51.5 |
| PA2067 | | probable hydrolase | -14.9 |
| PA2069 | | probable carbamoyl transferase | -11.3 |
| PA2119 | | alcohol dehydrogenase (Zn-dependent) | -18.4 |
| PA2193 | *hcnA* | hydrogen cyanide synthase HcnA | -32.0 |
| PA2204 | | putative binding protein component of ABC transporter | -78.0 |
| PA2331 | | hypothetical protein | -7.2 |
| PA2441 | | hypothetical protein | -7.8 |
| PA2501 | | hypothetical protein | -9.1 |
| PA2634 | *aceA* | isocitrate lyase AceA | -23.8 |
| PA2950 | *fabV* | FabV, proton motive force protein, PFM enoyl-acyl-carrier protein reductase | -10.5 |
| PA2968 | *fabD* | malonyl-CoA-[acyl-carrier-protein] transacylase | -12.6 |
| PA2970 | *rpmF* | 50S ribosomal protein L32 | -13.4 |
| PA2999 | *nqrA* | Na + -translocating NADH:ubiquinone oxidoreductase subunit Nrq1 | -10.5 |
| PA3001 | | probable glyceraldehyde-3-phosphate dehydrogenase | -28.4 |
| PA3162 | *rpsA* | 30S ribosomal protein S1 | -17.8 |
| PA3166 | *pheA* | chorismate mutase | -8.1 |
| PA3204 | *cpxR* | two-component response regulator CpxR | -11.5 |
| PA3309 | | conserved hypothetical protein | -13.4 |
| PA3313 | | putative periplasmic organophosphorus compound-binding protein | -10.3 |

*(Continued)*

**Table 1.** (Continued)

| | | Descriptions | TBCF10839 *rubB*::Tn5 |
|---|---|---|---|
| PA3326 | clpP2 | ClpP2 | -20.8 |
| PA3330 | | probable short chain dehydrogenase | -26.1 |
| PA3337 | rfaD | ADP-L-glycero-D-mannoheptose 6-epimerase | -26.0 |
| PA3397 | fpr | ferredoxin--NADP+ reductase | -42.3 |
| PA3441 | | probable molybdopterin-binding protein | -11.4 |
| PA3446 | | conserved hypothetical protein | -8.9 |
| PA3525 | argG | argininosuccinate synthase | -17.5 |
| PA3621 | fdxA | ferredoxin I | -20.8 |
| PA3625 | surE | survival protein SurE | -10.8 |
| PA3635 | eno | enolase | -12.1 |
| PA3700 | lysS | lysyl-tRNA synthetase | -11.6 |
| PA3724 | lasB | elastase LasB | -28.0 |
| PA3743 | trmD | tRNA (guanine-N1)-methyltransferase | -18.7 |
| PA3745 | rpsP | 30S ribosomal protein S16 | -33.9 |
| PA3806 | | conserved hypothetical protein | -8.4 |
| PA3807 | ndk | nucleoside diphosphate kinase | -11.2 |
| PA3815 | iscR | IscR, modulator of catalase activity | -16.0 |
| PA3834 | valS | valyl-tRNA synthetase | -23.3 |
| PA3904 | | PAAR4 | -13.7 |
| PA3906 | | co-chaperone, co-TecT | -8.4 |
| PA3907 | tseT | TOX-REase-5 domain-containing effector, TseT | -9.2 |
| PA4064 | | probable ATP-binding component of ABC transporter | -10.5 |
| PA4067 | oprG | outer membrane protein OprG precursor | -17.6 |
| PA4131 | | probable iron-sulfur protein | -38.0 |
| PA4133 | | cytochrome c oxidase subunit (cbb3-type) | -12.3 |
| PA4134 | | hypothetical protein | -26.8 |
| PA4139 | | hypothetical protein | -70.7 |
| PA4140 | | hypothetical protein | -34.0 |
| PA4141 | | hypothetical protein | -64.1 |
| PA4211 | phzB1 | probable phenazine biosynthesis protein | -14.3 |
| PA4234 | uvrA | excinuclease ABC subunit A | -8.7 |
| PA4235 | bfrA | bacterioferritin | -24.7 |
| PA4238 | rpoA | DNA-directed RNA polymerase alpha chain | -18.2 |
| PA4239 | rpsD | 30S ribosomal protein S4 | -17.4 |
| PA4240 | rpsK | 30S ribosomal protein S11 | -18.8 |
| PA4242 | rpmJ | 50S ribosomal protein L36 | -10.1 |
| PA4244 | rplO | 50S ribosomal protein L15 | -17.6 |
| PA4245 | rpmD | 50S ribosomal protein L30 | -18.3 |
| PA4246 | rpsE | 30S ribosomal protein S5 | -84.0 |
| PA4247 | rplR | 50S ribosomal protein L18 | -11.5 |
| PA4248 | rplF | 50S ribosomal protein L6 | -8.7 |
| PA4249 | rpsH | 30S ribosomal protein S8 | -10.1 |
| PA4253 | rplN | 50S ribosomal protein L14 | -9.3 |
| PA4254 | rpsQ | 30S ribosomal protein S17 | -14.1 |
| PA4256 | rplP | 50S ribosomal protein L16 | -12.9 |

*(Continued)*

**Table 1.** (Continued)

|  |  | Descriptions | TBCF10839 *rubB*::Tn5 |
|---|---|---|---|
| PA4257 | *rpsC* | 30S ribosomal protein S3 | -11.3 |
| PA4258 | *rplV* | 50S ribosomal protein L22 | -25.9 |
| PA4259 | *rpsS* | 30S ribosomal protein S19 | -28.8 |
| PA4260 | *rplB* | 50S ribosomal protein L2 | -12.4 |
| PA4261 | *rplW* | 50S ribosomal protein L23 | -18.7 |
| PA4262 | *rplD* | 50S ribosomal protein L4 | -47.4 |
| PA4263 | *rplC* | 50S ribosomal protein L3 | -54.4 |
| PA4268 | *rpsL* | 30S ribosomal protein S12 | -32.6 |
| PA4271 | *rplL* | 50S ribosomal protein L7/ L12 | -11.3 |
| PA4272 | *rplJ* | 50S ribosomal protein L10 | -11.5 |
| PA4274 | *rplK* | 50S ribosomal protein L11 | -8.4 |
| PA4328 |  | hypothetical protein | -16.3 |
| PA4338 |  | hypothetical protein | -9.0 |
| PA4348 |  | conserved hypothetical protein | -47.7 |
| PA4429 |  | probable cytochrome c1 precursor | -18.6 |
| PA4430 |  | probable cytochrome b | -17.6 |
| PA4432 | *rpsI* | 30S ribosomal protein S9 | -13.1 |
| PA4433 | *rplM* | 50S ribosomal protein L13 | -15.5 |
| PA4459 | *lptC* | Lipopolysaccharide export system protein LptC | -25.6 |
| PA4483 | *gatA* | Glu-tRNA(Gln) amidotransferase subunit A | -7.6 |
| PA4563 | *rpsT* | 30S ribosomal protein S20 | -36.1 |
| PA4569 | *ispB* | octaprenyl-diphosphate synthase | -11.9 |
| PA4572 | *fklB* | peptidyl-prolyl cis-trans isomerase FklB | -19.7 |
| PA4610 |  | hypothetical protein | -14.3 |
| PA4642 |  | hypothetical protein | -9.2 |
| PA4665 | *prfA* | peptide chain release factor 1 | -11.5 |
| PA4694 | *ilvC* | ketol-acid reductoisomerase | -40.6 |
| PA4696 | *ilvI* | acetolactate synthase large subunit | -17.6 |
| PA4705 | *phuW* | PhuW | -8.4 |
| PA4715 |  | probable aminotransferase | -10.4 |
| PA4740 | *pnp* | polyribonucleotide nucleotidyltransferase | -13.1 |
| PA4741 | *rpsO* | 30S ribosomal protein S15 | -14.6 |
| PA4744 | *infB* | translation initiation factor IF-2 | -11.8 |
| PA4757 |  | conserved hypothetical protein | -10.4 |
| PA4770 | *lldP* | L-lactate permease | -11.4 |
| PA5128 | *secB* | secretion protein SecB | -12.2 |
| PA5161 | *rmlB* | dTDP-D-glucose 4,6-dehydratase | -9.9 |
| PA5170 | *arcD* | arginine/ornithine antiporter | -19.8 |
| PA5274 | *rnk* | nucleoside diphosphate kinase regulator | -11.5 |
| PA5316 | *rpmB* | 50S ribosomal protein L28 | -14.3 |
| PA5322 | *algC* | phosphomannomutase AlgC | -10.5 |
| PA5446 |  | hypothetical protein | -52.4 |
| PA5455 |  | putative glycosyltransferase | -7.6 |
| PA5468 |  | probable citrate transporter | -13.0 |
| PA5553 | *atpC* | ATP synthase epsilon chain | -10.9 |

*(Continued)*

**Table 1.** (Continued)

| | | Descriptions | TBCF10839 *rubB*::Tn*5* |
|---|---|---|---|
| PA5554 | *atpD* | ATP synthase beta chain | -15.8 |
| PA5555 | *atpG* | ATP synthase gamma chain | -10.9 |
| PA5557 | *atpH* | ATP synthase delta chain | -54.2 |
| PA5558 | *atpF* | ATP synthase B chain | -13.7 |
| PA5559 | *atpE* | ATP synthase C chain | -9.2 |
| PA5570 | *rpmH* | 50S ribosomal protein L34 | -137.3 |
| tRNA_Ile | | tRNA_Isoleucine, 723696–723772 (+) strand | -16.8 |

consumption of NADH reduction equivalents. Thus, it saves resources and maintains a restricted lifestyle. The mutant moreover showed only low or very low levels of mRNA transcripts of genes relevant for transport, secretion, signalling and virulence, i.e., quorum sensing (QS), the synthesis of hydrogen cyanide and phenazines and the production of type VI secretion systems (T6SS) and their associated effectors. The *rsaL*, *lasI* and *pqsABCE* genes were consistently more strongly expressed in TBCF10839 than in TBCF10839 *rubB*::Tn*5* (median 15-fold; range 10- to 63-fold).

### A functional rubredoxin reductase is necessary for quorum sensing

The strong downregulation of transcripts of the quorum sensing circuit suggests that the production of *N*-acyl homoserine lactone (AHL) signal molecules is impaired in the rubredoxin reductase – deficient strain [34–36]. As shown in Fig 2, indeed no AHLs were detectable with TBCF10839 *rubB*::Tn*5*, whereas complementation *in trans* with the *rubB* – bearing plasmid restored the production of AHLs to wild type levels.

Macromolecules and structurally complex compounds are at least partially degraded in the extracellular milieu prior to cellular uptake. Thus, the *P. aeruginosa* cell has to secrete hydrolytic enzymes. The secretion of elastase is under the control of QS [34–36]. To test for this phenotype, TBCF10839, TBCF10839 *rubB*::Tn*5* and complemented TBCF10839 *rubB*::Tn*5*; pME6010::TB*rubB* were tested for their ability to grow on casein. Wild type and complemented mutant rapidly utilized casein, whereas TBCF10839 *rubB*::Tn*5* did not (Fig 2D). To quantify the phenotype, the elastinolytic activity mediated by pseudolysin and to lesser extent staphylolysin was assessed with the elastin - Congo red assay [37]. Wild type TBCF10839 degraded elastin as a measure of elastase activity after entering the stationary phase of growth as expected for protease expression under QS control, whereas the enzymatic activity of TBCF10839 *rubB*::Tn*5* was reduced (Fig 2E), consistent with the observation (Fig 2A–2C) that quorum sensing is impaired in the absence of an intact rubredoxin reductase gene.

### Rubredoxin reductase – rubredoxin confer oxidative stress protection

Insertion of a transposon into *rubB* sensitized *P. aeruginosa* TBCF10839 to killing by polymorphonuclear neutrophils (PMNs) (Table 2) [25,26,31]. TBCF10839 is a highly virulent strain that persists and multiplies within PMNs [26]. Since inactivation of *rubB* re-sensitized TBCF10839 to the bactericidal action of PMNs, we hypothesized that the rubredoxin reductase/ rubredoxin electron transfer system may inactivate oxygen intermediates. During oxidative burst PMNs produce hydrogen peroxide and convert hydrogen peroxide and chloride to hypochlorous acid [42].

To test the role of RubB in the stress response to hydrogen peroxide, the wild-type TBCF10839, the TBCF10839 *rubB*::Tn*5* mutant and the complemented mutant TBCF10839 *rubB*::Tn*5*; pME6010::TB*rubB* were grown for 16h in LB or M9 to stationary phase, pelleted, diluted to a cell density of $OD_{600nm}$ of 0.05 and exposed to 0.2 M $H_2O_2$ for 5min. Dropping

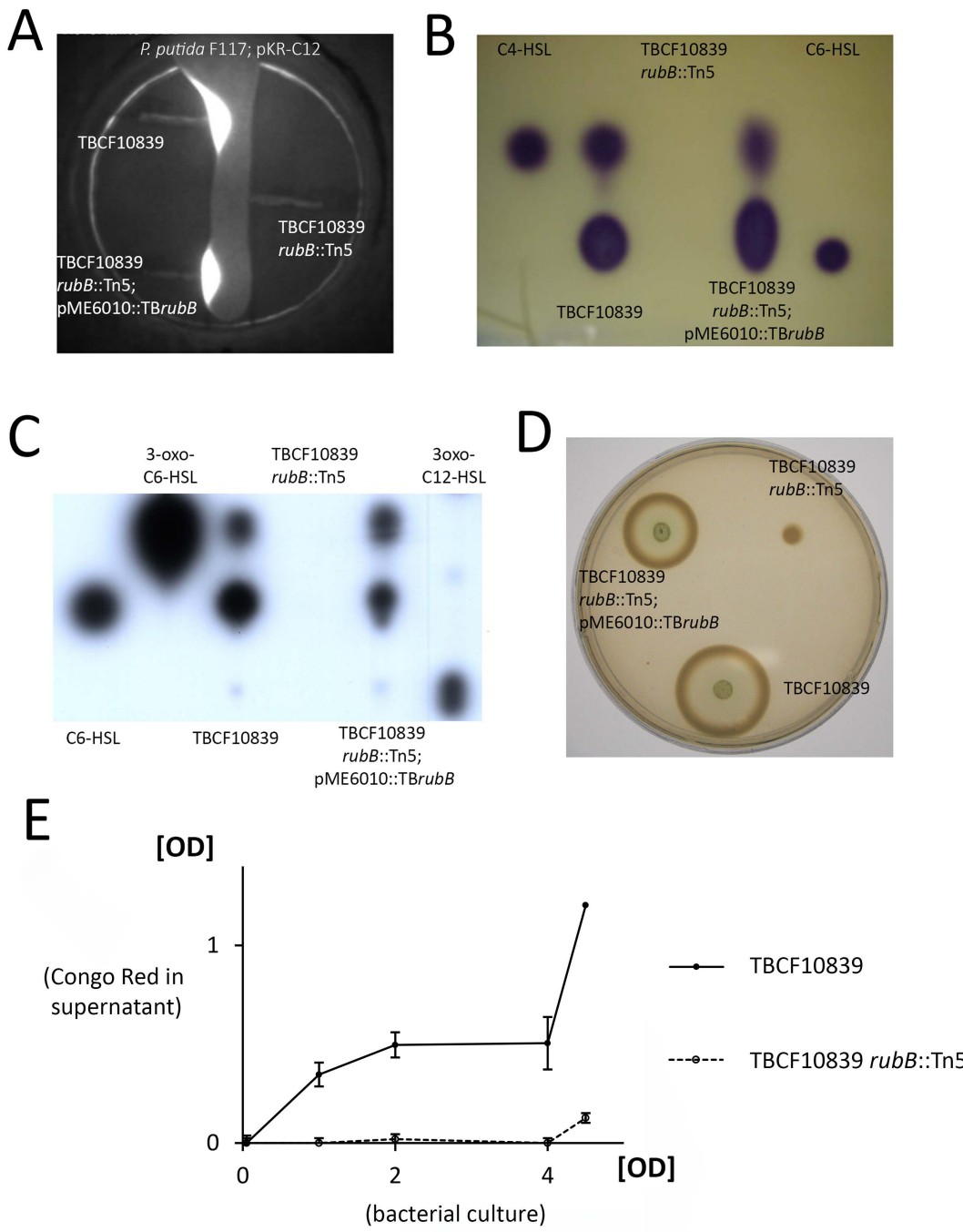

**Fig 2. Secretion of AHLs by *P. aeruginosa* TBCF10839, TBCF10839 *rubB*::Tn*5* and the complemented mutant TBCF10839 *rubB*::Tn*5*; pME6010::TB*rubB* [38]. A**. Tested *P. aeruginosa* strains were cultivated with *P. putida* F117 harboring plasmid pKR-C12, which contains a translational fusion of the *lasB* promoter to *gfp* and the *lasR* gene under control of a *lac*-type promoter [39]. Illumination with blue light excites green fluorescent protein. Please note the full restoration of the ability to produce AHLs in the complemented mutant. **B**. TLC analysis of AHLs secreted by tested *P. aeruginosa* strains using the AHL biosensor *Chromobacterium violaceum* CV026 [40], which is able to detect $C_4$- and $C_6$-HSL. **C**. TLC analysis of AHLs secreted by tested *P. aeruginosa* strains using AHL biosensor *E. coli* MT102(pSB403) [41], which is able to detect 3-oxo-$C_6$ and 3-oxo-$C_{12}$ HSLs. **A, B, C**. TBCF10839 and the complemented mutant TBCF10839 *rubB*::Tn*5*; pME6010::TB*rubB* secreted AHLs, whereas no detectable amounts of AHL were secreted in the three assays by the rubredoxin reductase knock out mutant TBCF10839 *rubB*::Tn*5* (n = 3 biological replicates). **D**. Proteolytic activity of

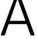

*P. aeruginosa*. Wild type TBCF10839 and the complemented mutant TBCF10839 *rubB*::Tn*5*; pME6010::TB*rubB* grew on casein, whereas casein could not be metabolized by TBCF10839 *rubB*::Tn*5* (> 20 independent assays). **E.** Absorption of Congo Red at 495 nm of the solubilization of elastin-Congo Red [37] in supernatants of *P. aeruginosa* TBCF10839 and TBCF10839 *rubB*::Tn*5* as a function of the optical density of the bacterial culture at 600 nm. Experiments were performed five times.

**Table 2. Phagocytosis assay. Intracellular *P. aeruginosa* after exposure to AB serum or PMNs for 90 min[a].**

| Ratio of *P. aeruginosa* CFUs in AB serum (opsonins): | |
| --- | --- |
| TBCF10839 *rubB*::Tn*5*/ TBCF10839 | 0.4 ± 0.1 |
| Ratio of intracellular *P. aeruginosa* CFUs in PMNs:[b] | |
| TBCF10839 *rubB*::Tn*5*/ TBCF10839 | 0.045 ± 0.009**** |
| TBCF10839 *rubB*::Tn*5*; pME6010::TB*rubB*/ TBCF10839 | 0.283 ± 0.043*** |
| TBCF10839 *rubB*::Tn*5*; pME6010::TB*rubB*/ TBCF10839 *rubB*::Tn*5* | 6.27 ± 1.47*** |

[a]In the phagocytosis test $10^7$ exponentially growing bacteria are exposed to $10^6$ freshly prepared human PMNs [25,31]. By time point 90 min about 100 viable *P. aeruginosa* bacteria were typically detectable in six clonally unrelated test strains [26], whereas 100- to 1000-fold more intracellular CFUs were isolated from the TBCF10839 strain (30 biological replicates from 10 human donors) [26].

[b]The data represent 10 biological replicates for each strain (mean ± SD; ***, $P < 10^{-3}$; ****, $P < 10^{-4}$; t-test).

10 µl aliquots showed large colonies of TBCF10839 and the complemented mutant, but none of TBCF10839 *rubB*::Tn*5* after overnight incubation. This assay demonstrated that the brief exposure to a lethal dose of hydrogen peroxide killed all RubB-deficient bacteria, but was survived by a fraction of the bacteria with a working RubB. Likewise, when LB supplemented with 10 mM $H_2O_2$ was inoculated with a cell density of $OD_{600nm}$ of 0.05, the liquid cultures became turbid (OD > 3) 48 h later if they had been inoculated with TBCF10839 or complemented mutant while no growth was observed in the case of TBCF10839 *rubB*::Tn*5* (Fig 3A).

**A**

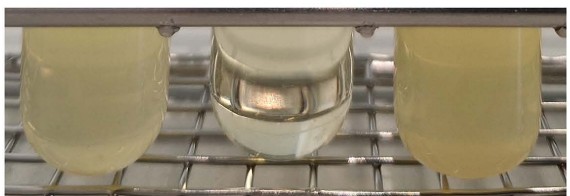

TBCF10839                    TBCF10839 *rubB*::Tn*5*;
                             pME6010::TB*rubB*

        TBCF10839 *rubB*::Tn*5*

**B**

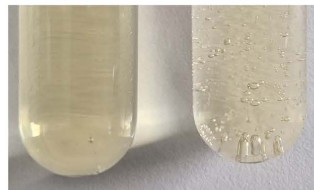

TBCF10839        TBCF10839 *rubB*::Tn*5*

**Fig 3. Exposure of *P. aeruginosa* to hydrogen peroxide. A.** TBCF10839, TBCF10839 *rubB*::Tn*5* and TBCF10839 *rubB*::Tn*5*; pME6010::TB*rubB* (in presence of 0.2 mg/mL tetracycline) were grown for 16 h in LB medium. Then a diluted cell suspension ($OD_{600nm}$ of 0.05) thereof was grown in LB supplemented at time zero with 10 mM $H_2O_2$. The figure shows the bacterial cell density after 48 h culturing at 37 °C (n = 2 biological replicates performed in triplicate). **B.** Acute exposure to 0.32 M $H_2O_2$. Within the first 30 seconds *P. aeruginosa* TBCF10839 (left) reduces hydrogen peroxide to water, whereas TBCF10839 *rubB*::Tn*5* (right) decomposes hydrogen peroxide by disproportion to produce water and oxygen foam. The figure shows typical examples of experiments that have been performed more than fifty times. To see the kinetics of a mixture of NADH, rubredoxin reductase, rubredoxin and catalase exposed to hydrogen peroxide in real time, the reader is referred to the video in the supplement.

To examine this presumed kinetic role of RubB in the stress response to hydrogen peroxide, the wild-type TBCF10838, the TBCF10839 *rubB*::Tn*5* mutant and the complemented mutant TBCF10839 *rubB*::Tn*5*; pME6010::TB*rubB* were grown for 16 h in LB to stationary phase, pelleted, washed with mineral medium and re-suspended in one tenth of the growth volume. The bacterial suspension was then exposed to a lethal dose of 3% (v/v) hydrogen peroxide. The transposon mutant, but neither wild-type nor complemented strain immediately produced bubbles of oxygen within the first ten seconds (Fig 3B). Thus, wild type and complemented strain initially reduced hydrogen peroxide by virtue of their rubredoxin reductase, whereas TBCF10839 *rubB*::Tn*5* degraded hydrogen peroxide by disproportionation into oxygen and water from the very beginning.

To quantify the impact of rubredoxin reductase for *P. aeruginosa* TBCF10839 to withstand hydrogen peroxide, bacteria were either grown in liquid culture (Fig 4A) or on plates (Fig 4B). TBCF10839 and TBCF10839 *rubB*::Tn*5* were grown in parallel for 10 h in LB with ten two-fold serial dilutions of initial concentrations of 128 mM to 0.25 mM $H_2O_2$. TBCF10839 *rubB*::Tn*5* could cope with $H_2O_2$ concentrations at and below 2 mM (Fig 4A). Conversely, TBCF10839 was continuously growing up to 8 mM and after an initial delay gained growth at all higher concentrations of up to 128 mM (Mann-Whitney test, $P = 5 \times 10^{-7}$) (Fig 4A). A portion of the bacterial inoculum of about $10^8$ CFU/ml survived the initial insult of high concentrations of hydrogen peroxide so that the bacterial community could expand when $H_2O_2$ was gradually degraded over time to lesser toxic concentrations. Isolated bacteria were more susceptible to killing by hydrogen oxide than the bacterial communities in liquid culture. When about 100 CFU of exponentially growing bacteria were plated onto a series of LB agar plates whereby the concentration of hydrogen peroxide was increased in steps of two-fold, colonies of TBCF10839 *rubB*::Tn*5* and TBCF10839 emerged up to concentrations of 0.2 and 0.8 mM $H_2O_2$, respectively (Fig 4B). The wild-type strain was more resistant to hydrogen peroxide than the mutant (log rank test, $P = 0.01$).

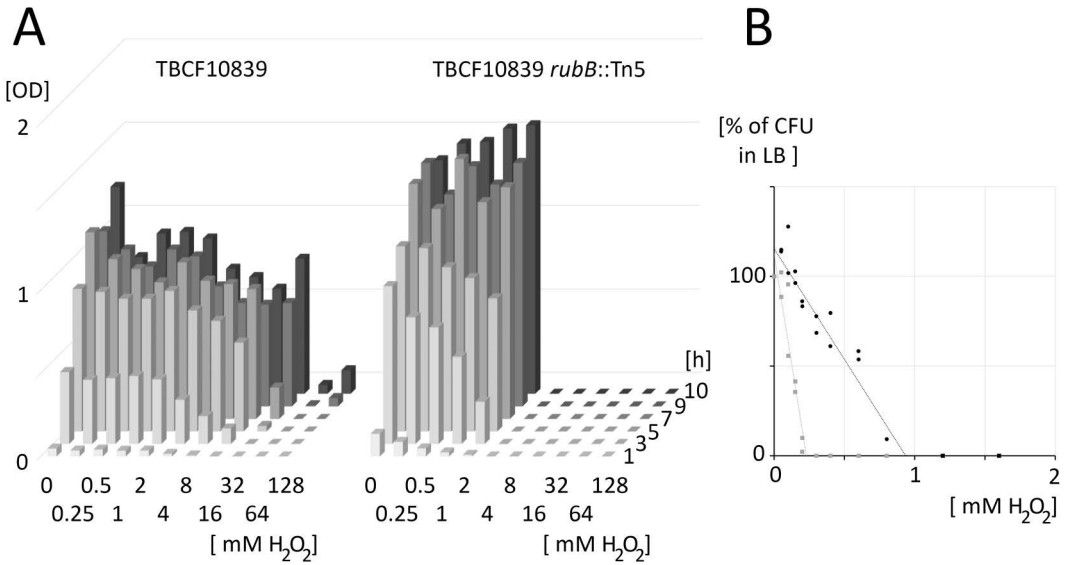

**Fig 4. Bacterial growth at 37 °C in liquid medium (A) or plates (B) during exposure to hydrogen peroxide. (A)** An inoculum of $OD_{600 \, nm}$ of 0.05 of TBCF10839 *rubB*::Tn*5* (right) and TBCF10839 (left) was grown in LB under shaking (120 rpm) during exposure to initial concentrations of 0.25 mM to 128 mM $H_2O_2$. **(B)** Normalized colony counts after plating about 100 CFU of exponentially growing TBCF10839 *rubB*::Tn*5* (gray squares) and TBCF10839 (black circles) onto LB agar plates containing two-fold dilutions of 2.4 mM or 3.2 mM $H_2O_2$. The linear decline of bacterial survival differed by 4-fold between TBCF10839 *rubB*::Tn*5* (slope: -49% killing per 0.1 mM $H_2O_2$, $R^2 = 0.88$) and TBCF10839 CFUs (slope: -12.3% killing per 0.1 mM $H_2O_2$, $R^2 = 0.87$). Experiments (A) and (B) were performed in duplicate with the same bacterial precultures.

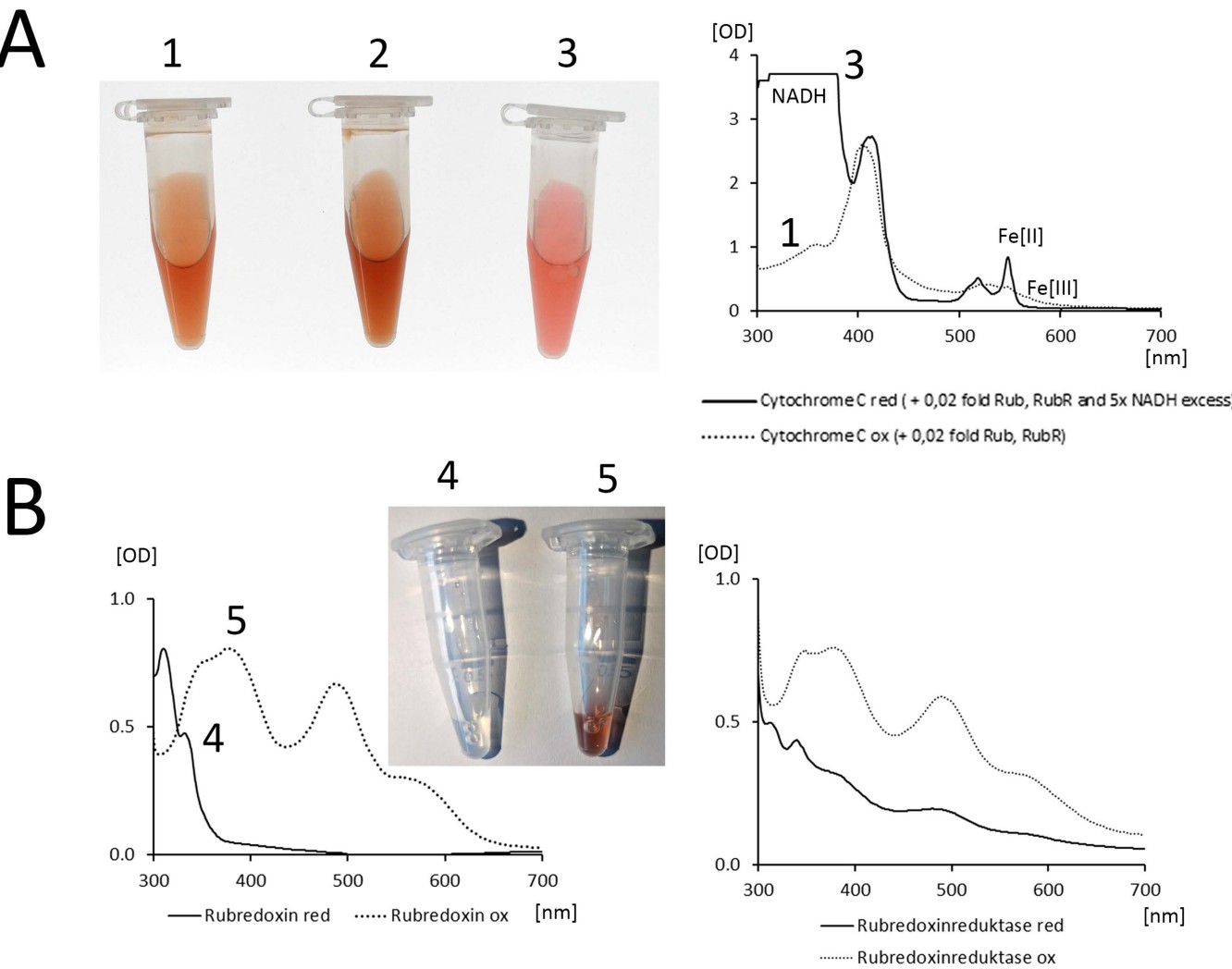

**Fig 5. A. Left:** The three Eppendorf tubes show PBS supplemented with 40 μM horse cytochrome C, (1), then supplemented with 4.1 μM rubredoxin RubA2 and 3.3 mM NADH (2) and finally with 3.1 μM *P. aeruginosa* rubredoxin reductase RubR (3) in PBS. **Right:** Absorption spectrum of 25 μM horse cytochrome C, 1.25 μM RubA2, 1.25 μM RubR (1, dotted line) plus 125 μM NADH (3, solid line).The NADH – RubR – RubA2 electron transfer system reduces the iron center of cytochrome C. **B.** Absorption spectra of reduced (solid line) and oxidized (dotted line) of 4.1 μM rubredoxin RubA2 (left) and 3.1 μM *P. aeruginosa* rubredoxin reductase RubR (right). The insert shows reduced (Eppendorf tube 4) and oxidized (Eppendorf tube 5) RubA2. Assays with Eppendorf tubes were repeated more than ten times, absorption spectra were repeatedly recorded at six different dates.

### *P. aeruginosa* rubredoxin reductase – rubredoxin complex is a fast and promiscuous electron transfer system

The rubredoxin reductase/ rubredoxin complex has so far primarily been examined as an electron transfer system for the alkane monooxygenase. According to recently published cryo-EM data rubredoxin reductase and alkane monooxygenase bind to the same surface of rubredoxin [19]. Thus, it seems reasonable that the rubredoxin reductase/ rubredoxin complex transfers electrons also to other acceptor proteins. Otherwise, its role in protection against $H_2O_2$ and killing by neutrophils would not be plausible. Hence, we first tested whether the recombinant rubredoxin reductase/ rubredoxin complex will transfer electrons to a phylogenetically distant iron center – bearing metalloprotein. Taking horse cytochrome C as an example, rubredoxin reduced the iron center of a member of a phylogenetically ancient protein family (Fig 5).

Next, we wanted to support our interpretation that catalase in the presence of an active rubredoxin reductase/NADH/rubredoxin complex will reduce $H_2O_2$ and thereby prevents the disproportionation into oxygen and water as long as the reduction equivalent NADH can be consumed. The video in the supplement compares the kinetics of oxygen production after exposure to $H_2O_2$ to a mixture of rubredoxin reductase, rubredoxin and catalase in the absence and presence of NADH. Whereas in the absence of NADH oxygen is immediately produced by disproportionation, the reaction is delayed and less intense in presence of an active rubredoxin reductase/rubredoxin complex. Hence, we concluded that the electron transfer from NADH to rubredoxin reductase/rubredoxin and then to catalase must be faster than the second step of the disproportionation reaction. In other words, the rubredoxin reductase/rubredoxin complex can transform the catalase to a highly efficient reductase.

**Kinetics of the Rubredoxin Reductase - mediated reduction of NADH monitored by stopped-flow experiments.** To verify this conclusion that the rubredoxin reductase/rubredoxin complex is faster than the second step of catalase-mediated reduction of $H_2O_2$, we measured the kinetics of electron transfer in rubredoxin reductase and the rubredoxin reductase/rubredoxin complex with the isolated recombinant proteins in stopped-flow instruments. The reaction was followed by monitoring the fluorescence emission of the intramolecular oxidized FAD isoforms of rubredoxin reductase. At the chosen excitation wavelength of 436 nm the reduced $FADH_2$ does not absorb so that we could discriminate the fluorescent isoforms of oxidized FAD and the FAD semiquinone intermediates from the reduced non-fluorescent $FADH_2$ [43]. Fig 6 shows that after the addition of NADH the FAD fluorophore was reduced within milliseconds to $FADH_2$. The rate constant for the intramolecular electron transfer from the NADH binding cavity to the FAD binding pocket was deduced from the initial linear decline of FAD fluorescence to be $(1.5 \pm 0.5) \times 10^7$ $M^{-1}s^{-1}$ (Fig 6A), which is equivalent or even higher than the rate constant for the second step of the catalase-mediated disproportionation [44]. If we added an eight-fold molar excess of NADH and of rubredoxin to rubredoxin reductase, the FAD fluorescence intermittently increased with a maximum at 0.1 s indicating that the single electron transfer to rubredoxin presumably takes place via the flavin semiquinone reaction intermediates [43] (solid line, Fig 6C, 6D). Within the one second time range the FAD semiquinone intermediates were populated in the presence of reduced rubredoxin because FAD receives two electrons from NADH and transfers only single electrons to rubredoxin.

## Role of rubredoxin reductase in virulence

Since QS regulates social behavior and virulence of *P. aeruginosa* [36], we tested whether the pathogenicity of TBCF10839 was attenuated in the AHL-deficient TBCF10839 *rubB*::Tn*5* mutant using a *C. elegans* infection model that has become standard to identify virulence factors of *P. aeruginosa* [45]. By applying the Fast Killing model, synchronized *C. elegans* survived nearly as well with TBCF10839 *rubB*::Tn*5* as with their food source *Escherichia coli* DH5α, indicating that the inactivation of the rubredoxin reductase gene abrogated toxin-mediated death (Fig 7). Complementation of TBCF10839 *rubB*::Tn*5* with the recombinant plasmid pME6010::TB*rubB in trans* restored the fast killing kinetics, which was indistinguishable from that of wild type TBCF10839.

## Genome sequencing

By the end of our investigation, we performed deep sequencing of aliquots of stock cultures of the TBCF10839 and TBCF10839 *rubB*::Tn*5* strains that had been repeatedly used during this study. The genomes derived from the four tested stocks differed from the deposited sequence of TBCF10839 (accession no. PRJNA975170) by the missense variant p.Gly231Arg of the type VI secretion system effector protein TseT (S2 Table). The transposon mutant moreover carried a synonymous substitution in the *rlmF* gene encoding the 23S rRNA (adenine(1618)-N(6))-methyltransferase (S2 Table). These findings strongly indicate that our reported results are not biased by any secondary mutations that emerged elsewhere in the genome during the course of our study. Interestingly, deep sequencing unraveled a further 34 pairs of

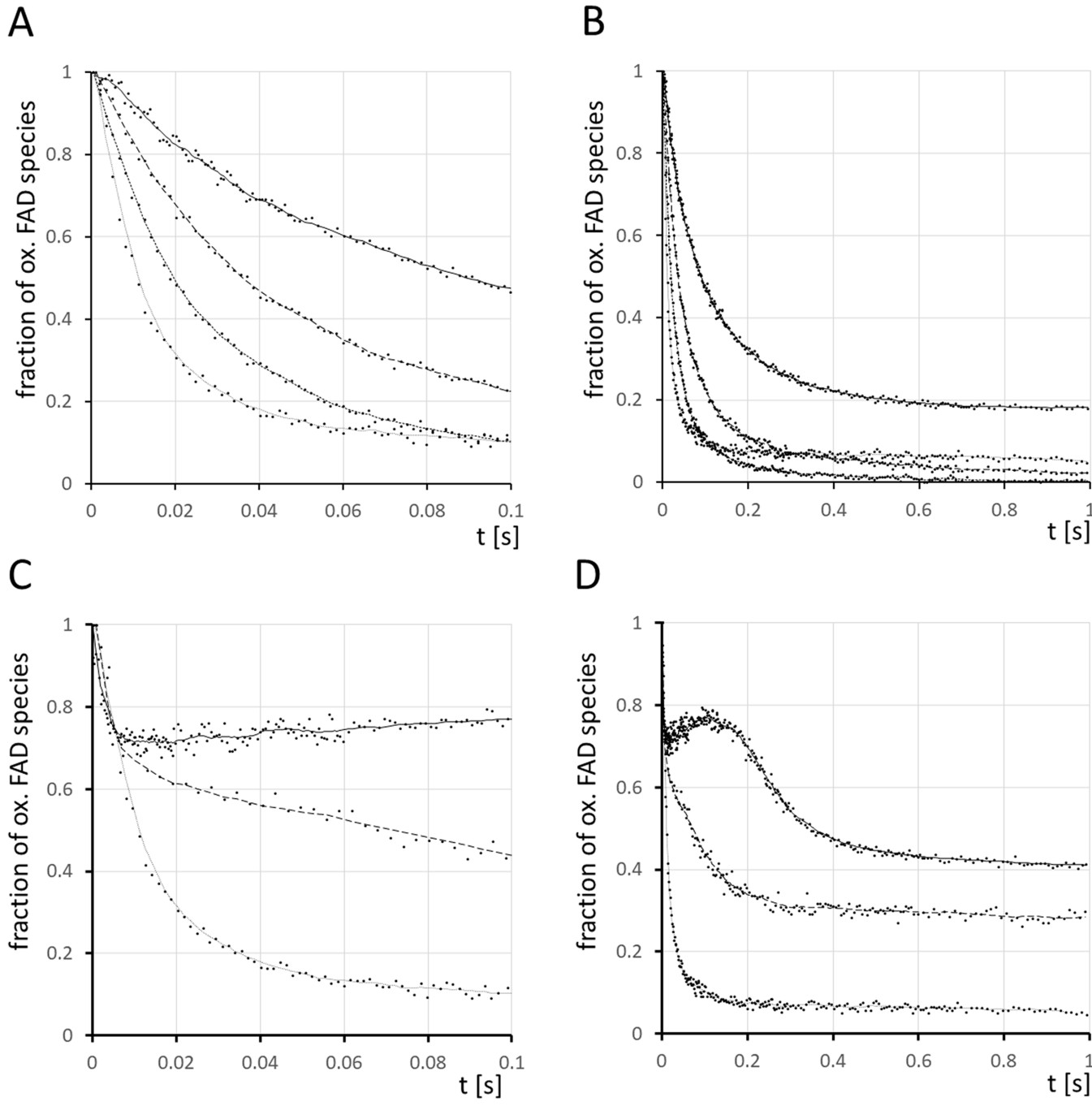

**Fig 6. Stopped-flow experiments of the kinetics of the transfer of reduction equivalents from NADH in bulk solution to the FAD prosthetic group of *P. aeruginosa* rubredoxin reductase RubR (upper panel) followed by the transfer of reduction equivalents of the FADH$_2$ intermediate of RubR to *P. aeruginosa* rubredoxin RubA2 (lower panel). A, B**. Equal volumes of 80 µL of 10 µM RubR were mixed with (from top to bottom) 80 µL 10 µM NADH (solid line), 20 µM NADH (long dashed line), 40 µM NADH (short dashed line), 80 µM NADH (dotted line) and the emitted fluorescence of flavin mononucleotide isoforms [43] was monitored through a Schott GG455 filter for the time period of 0 − 0.1 s (**A**) and 0 − 1 s (**B**). **C, D**. Equal volumes of 80 µL of 10 µM RubR were mixed with (from top to bottom) 80 µL 80 µM NADH/ 80 µM RubA2 (solid line), 80 µM NADH/ 20 µM RubA2 (long dashed line), 80 µM NADH/ 0 µM RubA2 (dotted line) and the emitted fluorescence of flavin mononucleotide isoforms was monitored through a Schott GG455 filter for the time period of 0 − 0.1 s (**C**) and 0 − 1 s (**D**). Altogether 314 experiments were performed with at least five technical replicates per condition.

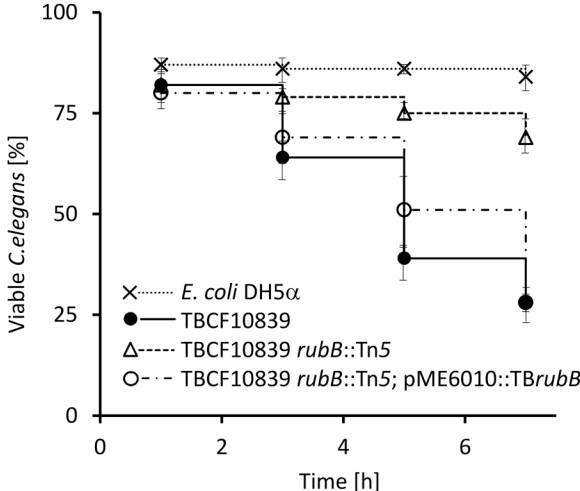

**Fig 7. Kaplan-Meier plot of the killing of *Caenorhabditis elegans* by *P. aeruginosa* TBCF10839 (black solid line), its isogenic TBCF10839 *rubB*::Tn*5* transposon mutant 41D3 (dashed line) and the complemented mutant TBCF10839 *rubB*::Tn*5*; pME6010::TB*rubB* (dot-dashed line).** Fifty L4 larvae were placed in each well and scored for dead worms by microscopic examination. *E. coli* DH5α served as a negative control (dotted line). Values are the mean±SD of two biological replicates assayed in triplicate. Survival rates of worms were equally low in the presence of TBCF10839 or TBCF10839 *rubB*::Tn*5*; pME6010::TB*rubB* ($P=0.51$, effect size 0.03), but significantly higher in the presence of TBCF10839 *rubB*::Tn*5* ($P<10^{-5}$, effect size 0.50).

sequence variants showing various proportions of the canonical and of another nucleotide at the respective genome position (S2 Table). Eight pairs of co-existing variants were shared by two or more stocks and the other 26 pairs were detected in only one stock. Twenty-six nucleotide pairs (76%) of which 16 pairs segregated in four 2- to 20-bp large hot spots (S2 Table), were located in the non-coding region that makes-up about 20% of the genome sequence [28]. Most nucleotide pairs (80%) represent C-G or G-C transversions (S2 Table) that apparently drive the on-going molecular evolution of the non-coding region of the *P. aeruginosa* TBCF10839 genome.

## Discussion

The function of the rubredoxin reductase/ rubredoxin complex has so far been mainly discussed in the context of the utilization of alkanes as a carbon source for bacteria [18,19,21]. The cytosolic rubredoxin reductase and rubredoxin shuttle electrons to the integral membrane protein AlkB that catalyzes the hydroxylation of the terminal C-H bond of alkanes using molecular oxygen. The generated alcohol is then stepwise oxidized to aldehyde and carboxylic acid, acetylated with acetyl CoA and fed into the cycle of fatty acid oxidation. While most bacteria encode the hydroxylase and the electron transfer system in separate genes, some fusion proteins have been identified [46,47]. Either rubredoxin is C-terminally fused to AlkB or ferredoxin and ferredoxin reductase are N-terminally fused to AlkB [17,18]. Such two-domain or three-domain protein structures should specify the electron transfer to AlkB and thus confine the substrate to alkanes. However, as shown here for *P. aeruginosa* TBCF10839, if rubredoxin and its reductase genes exist as separate entities, the corresponding proteins may fulfill multiple roles in the bacterial cell.

*P. aeruginosa* TBCF10839 engages rubredoxin reductase and rubredoxin for protection against oxidative stress. Upon exposure to a lethal dose of hydrogen peroxide, the TBCF10839 strain significantly changed the expression of about a quarter of its genes at the transcriptional level whereby catalase, cytochrome C oxidase and alkylhydroperoxide reductase mRNA transcripts were more than a hundredfold increased [48]. However, as shown in this report, it were not these enzymes but rubredoxin reductase and rubredoxin, which initially inactivated the insult by hydrogen peroxide within the

first 30 seconds and let the bacteria survive. The wild type strain also produced oxygen bubbles after about a 30 seconds delay, although at lower frequency than the mutant (see video in the supplement). Catalase is a heme $Fe^{3+}$ enzyme that converts hydrogen peroxide to water and oxygen in a two-step process [44,49,50]. During the first reaction, catalase reduces one molecule of hydrogen peroxide to water, thereby generating an oxoferryl porphyrin cation $Fe^{4+}$ radical [44,49]. Subsequently this high-valent iron intermediate, known as compound I, reacts with a second molecule of $H_2O_2$ to generate $O_2$ and a water molecule. We hypothesize that within the first seconds of the exposure to hydrogen peroxide, the rubredoxin reductase/rubredoxin complex continuously regenerates the $Fe^{3+}$ state of the catalase and prevents the inactivation of catalase by hydroxyl radicals to compound III that occurs at a thousand-fold lower rate than the electron transfer in the RubB/RubA2 complex [49]. Hence, the catalase remains active and initially will only execute the first reaction. As shown in our stopped-flow experiments with NADH and rubredoxin reductase or the rubredoxin reductase/rubredoxin complex, the redox process occurs at a similar or higher rate than the catalase-mediated disproportionation to oxygen and water [44]. In summary, the rubredoxin reductase/rubredoxin complex makes the catalase more efficient and moreover protects the enzyme from the deleterious production of the inactive compound III [44,49].

Electron transfer from NAD(P)H to rubredoxin reductase involves a reductive and an oxidative step with respect to rubredoxin [21]. NAD(P)H binding initiates the reductive half reaction, involving hydride transfer from NAD(P)H-C4 to FAD-N5 resulting in a blue charge transfer complex between $FADH^-$ and $NAD^+$. During the oxidative half reaction, two electrons are sequentially transferred from FADH to two rubredoxin molecules, resulting in a flavin semiquinone reaction intermediate [51]. By superimposing the structures of the recently resolved alkane hydroxylase – rubredoxin complex of *F. thermophila* [19] on the rubredoxin reductase/ rubredoxin complex of *P. aeruginosa* [21], Chai and colleagues [19] observed that the hydroxylase and the reductase bind to the same face on rubredoxin. Thus, rubredoxin must dissociate from the hydroxylase to associate with the reductase to enable its reduction and subsequently transfers the electron to the hydroxylase. This shuffling of rubredoxin between reductase and hydroxylase may explain our experimental finding that the rubredoxin reductase/ rubredoxin complex is promiscuous in *P. aeruginosa* TBCF10839 and serves as an electron donor for many more proteins than just the alkane hydroxylase. If the alkane hydroxylase can be substituted by many enzymes, the rubredoxin reductase/rubredoxin electron transfer system can become a global player to battle oxidative stress. This global role of rubredoxin reductase/rubredoxin to control the oxidation state of the iron centers of the *P. aeruginosa* cell is clearly visible to the naked eye by the different color of the cell pellets of wild type strain and its isogenic mutant with an inactivated rubredoxin reductase (Fig 1B).

Electron transfer by rubredoxin reductase/rubredoxin is not confined to endogenous substrates. As shown in Fig 5, the rubredoxin reductase/rubredoxin complex efficiently reduced mammalian cytochrome C. Likewise, the diffusible signal molecules of quorum sensing were not produced when *rubB* was inactivated by transposon insertion. The QS circuitry that controls motility, biofilm formation and production of virulence factors [36] was defective in TBCF10839 *rubB*::Tn*5* mutant. Virulence was attenuated, which was consistently observed in the worm infection model.

In summary, the rubredoxin reductase/ rubredoxin complex turned out to be a key player for physiology and virulence – at least for a highly pathogenic strain such as TBCF10839. The versatile and irreplaceable roles of rubredoxin reductase/ rubredoxin explain why the transposon mediated inactivation of *rubB* abrogated the strain-specific feature of TBCF10839 to persist and grow in neutrophils, man's major antipseudomonal weapon [2,26,52]. Since *rubAB* genes are widespread among microorganisms [17,18], the versatile role of the electron transfer system beyond alkane oxidation should not be confined to a few strains or taxa. It probably has not been studied in another context because rubredoxin and the reductase were first identified as canonical electron-transfer partners of AlkB in *P. putida* GPo1, the bacterium best characterized in liquid alkane metabolism [53–56]. However, parallel to this work, Stokas and colleagues have reported that the *alkK – rubA – rubB* operon is instrumental for intracellular replication of *Mycobacterium tuberculosis* in macrophages. If the *alkK – rubA – rubB* locus was overexpressed, the bacteria survived better than the wild type in macrophages [57]. Likewise, numerous clones of *P. aeruginosa* can survive in macrophages that are deficient in the cystic fibrosis-causing

gene *CFTR* [4,58]. Thus, the role of the rubredoxin reductase/rubredoxin complex for intracellular survival in defense cells of the mammalian host has now been demonstrated in two phylogenetically unrelated human pathogens.

## Materials and methods

### Bacterial strains

*P. aeruginosa* TBCF10839 was isolated from respiratory secretions of an individual with cystic fibrosis (CF) who was regularly attending the CF clinic of Hannover Medical School [26,59]. First subcultures were maintained in lysogeny broth (LB) supplemented with 15% (w/v) glycerol at −80 °C until use.

The *P. aeruginosa* TBCF10839 'signature tagged mutagenesis' (STM) transposon library was constructed with the plasposon pTnModOGm [60] carrying variable signature tags as described previously [25,61]. Auxotrophic mutants had been counterselected by growth of the transposon mutants on minimal medium with glycerol as the single carbon source. *P. aeruginosa* or *Escherichia coli* strains were routinely grown overnight as shaken cultures (230 rpm) in LB at 37 °C. Five mL LB was inoculated with a toothpick of frozen bacterial stock solution and incubated for 16–48 h. Recombinant *E. coli* DH5α strains transfected with pME6010 [62] were growing in the presence of 50 µg ml$^{-1}$ tetracycline and recombinant *P. aeruginosa* was cultured in the presence of 200 µg ml$^{-1}$ tetracycline.

### DNA preparation

*P. aeruginosa* genomic DNA was prepared from cells grown in LB following a protocol optimized for Gram-negative bacteria [63].

### Screening of the TBCF10839 'signature tagged mutagenesis' transposon library

The screening of the STM library by phagocytosis assay with granulocytes and the identification of attenuated mutants by hybridization of signature tags amplified by PCR of genomic DNA of the bacteria from the infection experiments onto the dot blot of all oligonucleotide tags were performed as described previously [31]. Plasmid rescue was performed to transfer the minitransposon with its flanking sequences as stable episomal plasmids into *E. coli* DH5α. The protocol of Dennis and Zylstra [60] was modified as follows: 10 µg *P. aeruginosa* DNA was digested with 40 U PstI overnight at 37 °C in 40 ml restriction buffer, purified by phenol/chloroform extraction, and the pellet was suspended in 25 µl TE buffer. An aliquot of 500 ng restricted genomic DNA was incubated with 1000 cohesive end ligation units of T4-DNA ligase for 6 h at 16 °C in a total volume of 25 µl ligase buffer. Forty to sixty ng ligated DNA were transformed into *E. coli* DH5α and plasmid-harboring cells were selected with gentamicin (30 µg/ml) on LB agar. The plasmids were used for sequencing of the genomic insertion site of the transposon.

Complementation of TBCF10839 *rubB*::Tn*5* was performed with the shuttle vector pME6010 (GenBank accession no. AF118810) that contains the 'RK2 tetracycline resistance protein' TetA and the 'RK2 tetracycline repressor protein' TetR of the Birmingham-group IncP-1α conjugative plasmid RK2 isolated from *Klebsiella aerogenes* K8841 as selectable markers [62]. DNA fragments containing the *rubB* gene were amplified from the genomic DNA of *P. aeruginosa* TBCF10839 and ligated into the BglII- and EcoRI-site of the shuttle vector pME6010 [62]. Introduction of the plasmid construct into strain TBCF10839 *rubB*::Tn*5* was carried out by electroporation. Complemented mutants were selected on LB agar containing 200 µg ml$^{-1}$ tetracycline. Sequencing of the PCR products was performed by Qiagen (Hilden, Germany).

### Whole genome sequencing

Fragment libraries were made of 50 ng genomic DNA with the NEBNext Ultra II DNA Library Prep Kit for Illumina (New England Biolabs, Frankfurt am Main, Germany; #E7645L) and NEBNext Multiplex Oligos for Illumina (#E6440) in six PCR cycles. Purification of fragment libraries was performed with 0.9 × AMPure Beads (Beckman Coulter, Krefeld, Germany;

#A63881). The Illumina NextSeq550 platform was applied for paired-end 2x 75 bp short-read sequencing (Illumina, Cambridge, UK; #20024904). Sequencing reads were aligned to the TBCF10839 reference genome (GenBank: CP127016.1) with BWA (version 0.7.17) [64]. The Genome Analysis Toolkit (GATK version 4.2.5.0) was used to perform variant calling [65].

## RNA isolation and GeneChip microarray analysis

Bacterial cells were harvested by centrifugation at $3,800 \times g$ for 2 min at 4 °C. Total RNA from approximately $3 \times 10^{10}$ cells was extracted with a modified hot phenol method as described previously [27]. The generation of cDNA and subsequent biotin-ddUTP terminal-labeling steps were performed as described in the manufacturer's instructions for the *P. aeruginosa* GeneChip (cat.no. 900339, Applied Biosystems, Thermo Fisher Scientific), using the 10 µg of total RNA mixed with random primers (Thermo Fisher Scientific) and control in vitro transcripts of 10 non-*Pseudomonas* gene sequences. GeneChip hybridization and washing were carried out following the manufacturer's instructions (Applied Biosystems, Thermo Fisher Scientific) and as described previously [48] using the Affymetrix Microarray Suite software (version 5.0) with Affymetrix default parameters. The average microarray hybridization signal intensity was scaled to 150. S1 Table in the Supporting Information shows the complete datasets and the numerical outcome of the comparison of the GeneChip hybridization signals. Two GeneChips for each strain per condition were compared by the four-comparison survival method [66] to search for genes that significantly changed their signal intensities by the Wilcoxon rank test, with a minimum of a twofold change in all four comparisons. The arithmetic average and the standard deviation of the four comparisons were calculated. As an independent criterion for significantly changed signal intensities, a Bonferroni correction of the signal ratios was applied to account for the number of tests, which in this case was the total number of 5,900 open reading frames (ORFs) on the chip. First, the ratio of calibrated and corrected hybridization signals per gene ($S_i$) obtained from cultures grown under identical conditions was verified to follow a Gaussian distribution, and the variance ($\sigma$) was calculated. mRNA transcript levels of a gene ($i$) were considered to be significantly differentially expressed, if the ratio $S(i)_A/S(i)_B$ or $S(i)_B/S(i)_A$ exceeds the threshold ($1 + u\sigma$), whereby the factor $u$ defines that upper boundary of the normalized Gaussian integral $\Phi(u)$ where $\Phi(u) = x^n$ matches the Bonferroni-corrected 95% confidence interval in the expression $(1 - \alpha) = x^n$ (here, n = 5,900, α = 0.025, and $0.975 \ll x < 1.0$). In summary, changes were only classified as significant if they fulfilled the criteria of the four-comparison survival method and exceeded the threshold of the Bonferroni correction for multiple testing.

## Phagocytosis assay

The isolation of neutrophils and the subsequent phagocytosis assay with *P. aeruginosa* bacteria were performed as described previously [26,31].

## Plate and culture assays

Mineral medium consisted of 0.04 g/L $CaCl_2$ x 6 $H_2O$; 20 mg/L $CuSO_4$; 2.8 mg/L $FeSO_4$ x 7 $H_2O$; 3 g/L $KH_2PO_4$; 0.49 g/L $MgSO_4$ x 7 $H_2O$; 20 mg/L $MnCl_2$ x 4 $H_2O$; 4 g/L $Na_2HPO_4$; 10 mg/L $Na_2MoO_4$ x 2 $H_2O$; 3 g/L NaCl; 2 g/L $(NH_4)_2SO_4$; 20 mg/L $ZnSO_4$ x 7 $H_2O$. To test protease secretion, mineral medium agar plates were supplemented at pH 6.8 with 0.8% (w/v) casein. Growth of colonies of wild-type TBCF10838, TBCF10839 *rubB*::Tn5 and TBCF10839 *rubB*::Tn5; pME6010::T-B*rubB* was examined at 37 °C for up to 24 h side-by-side on the same plate. Plate assays were performed in triplicate on at least two different occasions. To test the growth of the strains on hexadecane, hexadecane was provided as vapor phase. The plates contained mineral medium and 0.2% agar and no carbon source, and a filter paper embedded with 0.2 ml of hexadecane was fixed on the Petri dish cover. To test the oxidative stress response, an inoculum of bacteria ($OD_{600} = 0.05$) was cultured in LB (37 °C, 120 rpm, 10 h) in a series of two-fold dilutions from initial 128 mM to 0.025 mM hydrogen peroxide. Alternatively, about 100 CFU of an exponentially growing culture were plated on LB agar that at the time of inoculation of bacteria contained hydrogen peroxide within the concentration range of 0.05 mM to 3.2 mM.

### Elastin-Congo Red assay

Elastinolytic activity was determined by the elastin – Congo red assay as described by Kessler and Safrin [37].

### Expression and purification *of* rubredoxin reductase RubB and rubredoxin Rub*A*2

Recombinant RubB and RubA2 were expressed as His$_6$ fusion proteins as described previously [21]. In brief, proteins were produced in *Escherichia coli* Tuner Competent Cells -Novagen (cat.no. 70623, Sigma-Aldrich) in LB and 37 µg/ml chloramphenicol. At an OD$_{600}$ of 1.0 (37 °C), protein expression was induced by 0.5 mM isopropyl-β-d-thiogalactoside and continued overnight at 20 °C. Cells were centrifuged, resuspended, and lysed by French press, and cell debris was removed by centrifugation. Purification involved Ni-nitrilotriacetic acid affinity chromatography (Qiagen), anion exchange chromatography (MonoQ; GE Healthcare, Chalfont St. Giles, U.K.), and gel filtration (Superdex 75; GE Healthcare). Ni$^{2+}$-(because of Ni-nitrilotriacetic acid) and Fe$^{3+}$-binding RubA2 were separated during ion-exchange chromatography. Proteins were dialyzed against 100 mM NaCl, 50 mM Tris·HCl (pH 8). A 5 mM concentration of β-mercaptoethanol was added to RubB to prevent cysteine oxidation.

### Competition between rubredoxin reductase/ rubredoxin and catalase to inactivate H$_2$O$_2$ (video in supplement S3 Video)

The 0.1 ml reaction mixtures A (left) and (B) (right) consisted of 3.1 µM RubB, 4.1 µM RubA2, 1 mU catalase from *Corynebacterium glutamicum* (cat.no. 11668153108, CustomBiotech, Roche) in phosphate buffered saline, pH 7.4 at room temperature. Then NADH (3.3 mM final concentration) was added to solution (B). Thereafter 10 µL 0.2 M hydrogen peroxide (equivalent to 15 mM in reaction mixture) were added. The video shows the emergence of oxygen bubbles for a time period of 1 min.

### Kinetic experiments in vitro

Stopped-flow measurements were performed at 25 °C in a modified version of the Durrum-Gibson stopped-flow apparatus [67]. Syringes M and N were filled with either 10 µM RubB in 0.1 M Na$_2$HPO$_4$/NaH$_2$PO$_4$ (pH 8.0), 0.1 M NaCl buffer (M) and variable concentrations of 10, 20, 40 or 80 µM NADH in 0.1 M Na$_2$HPO$_4$/NaH$_2$PO$_4$ (pH 8.0), 0.1 M NaCl (N). Aliquots of 80 µl of solution (M) and (N) were mixed for each reaction allowing for 4 or more technical replicates. The fluorescence of the coenzyme FAD of RubB was excited at 436 nm and emission was monitored through a Schott GG455 filter thereby blocking any excitation and emission of NAD/NADH. At the excitation wavelength of 436 nm oxidized FAD and semiquinone intermediates absorb, but not the reduced FADH2 [43]. Fluorescence was recorded within time intervals of 1 ms up to 1 s with a computer equipped with a DiSys PCI analog/ digital input/ output card. To study the ternary reaction between RubB and RubA2 in the presence of NADH, syringe (N) contained 80 µM NADH and variable concentrations of 0, 10 or 80 µM RubA2 in 0.1 M Na$_2$HPO$_4$/NaH$_2$PO$_4$ (pH 8.0), 0.1 M NaCl.

### *N*-acylhomoserine lactone analysis

For analysis of the *N*-acylhomoserine lactones (AHLs) produced by *P. aeruginosa* we employed different biosensors in combination with thin-layer chromatography (TLC) [38]. 250 ml spent supernatants from *P. aeruginosa* cultures grown to an OD$_{600}$ of 1.0 were extracted twice with dichloromethane (250:100 supernatant/dichloromethane). The combined extracts were dried over anhydrous magnesium sulfate, filtered, and evaporated to dryness. Residues were dissolved in 250 µl ethyl acetate. 10-µl samples were then applied to C$_{18}$ reversed-phase TLC plates (Merck No. 115389) and were separated by using methanol (60% v/v) in water as the solvent. For detection of AHLs the TLC plate was overlaid with soft agar seeded with the *luxAB* based AHL biosensor *E. coli* MT102 (pSB403) (detects long-chain and short-chain AHLs; [41]) or biosensor *Chromobacterium violaceum* CV026 (high sensitivity for short-chain AHLs; [40]). AHLs present in the culture

extracts were identified by comparison with the mobilities ($R_f$-values) of the co-migrating reference compounds. Alternatively, *Pseudomonas putida* F117(pKR-C12), which is sensitive for 3-oxo-C$_{12}$-HSL, and the test strains were streaked onto the plates to form a "T" [39].

### Fast killing of *Caenorhabditis elegans*

*C. elegans* Bristol N$_2$ (wild type), provided by the *Caenorhabditis* Genetics Centre (University of Minnesota, Minneapolis, MN, USA), was maintained on nematode growth medium with *E. coli* DH5α as a food source. A total of $1.5 \times 10^3$ *P. aeruginosa* cells in 100 µl were plated (1% (w/v) peptone, 1% (w/v) NaCl, 1% (w/v) glucose, 0.15 M sorbitol, 1.7 (w/v)% agar) and incubated overnight at 37 °C. Fifty synchronized *C. elegans* nematodes (stage N4) were added onto the plates and the number of dead worms was counted after 1, 3, 5 and 7 h. Plates inoculated with *E. coli* DH5α served as negative controls. Killing curves were evaluated by Kaplan-Meier Survival Analysis (log rank test using the chi-square distribution).

## Supporting information

**S1 Table. Gene chip microarray datasets: Comparison of TBCF10839 and the TBCF10839 *rubB*::Tn*5* mutant grown aerobically side-by-side in ABC minimal medium (two independent biological replicates each).** S1 Table lists the microarray hybridization data and their subsequent evaluation by the four-comparison survival method. (XLSX)

**S2 Table. Genome sequencing of aliquots of four stock cultures of *P. aeruginosa* TBCF10839 and TBCF10839 rubB::Tn*5* (long-term storage at -80 °C): Identification of single nucleotide variants.** (XLSX)

**S3 Video. Modification of the catalase-mediated degradation of hydrogen peroxide in the presence of reduced rubredoxin – rubredoxin reductase complex.** Both reaction vials contain the same amounts of 3.1 µM *P. aeruginosa* rubredoxin reductase RubB (RdxR), 4.1 µM *P. aeruginosa* rubredoxin RubA2 (Rdx) and 1 mU *Corynebacterium glutamicum* catalase in 80 µL PBS. By video time 5 s the RubB/RubA2 complex in the right vial is reduced with 3.3 mM NADH. By video time 40 s 15 mM hydrogen peroxide are added to the left vial (RubB/RubA2, catalase without NADH). Oxygen bubbles become visible by video time 44 s. By video time 60 s 15 mM hydrogen peroxide are added to the right vial (RubB/RubA2, catalase with NADH). Few small oxygen bubbles emerge by video time 80 s. Please note the substantially lower number and later appearance of smaller bubbles in the right vial by the end of the observation period. After the addition of hydrogen peroxide the catalase mediated disproportionation is slowed by about tenfold in the presence of the reduced RubA2/RubB complex. Thus, catalase initially acts as a reducing peroxidase as long as reduced RubA2 is available. The reaction kinetics shown in this video has been recorded in more than ten independent experiments. (MP4)

## Acknowledgments

We cordially thank Lisa Aberfeld, Marie Dorda, Ina Funk, Simon Genovese, Birgit Huber, Marie Kümmel, Ulrike Laabs, Kathrin Riedel and Susanne von Pall de Tolna for experimental support.

## Author contributions

**Conceptualization:** Lutz Wiehlmann, Burkhard Tümmler.

**Formal analysis:** Lutz Wiehlmann, Thorsten Adams, Sonja Horatzek, Burkhard Tümmler.

**Funding acquisition:** Burkhard Tümmler.

**Investigation:** Lutz Wiehlmann, Thorsten Adams, Claus Urbanke, Sonja Horatzek, Doris Jordan, Ilona Rosenboom, Burkhard Tümmler.

**Methodology:** Lutz Wiehlmann, Thorsten Adams, Claus Urbanke, Sonja Horatzek, Doris Jordan, Ilona Rosenboom, Leo Eberl, Burkhard Tümmler.

**Project administration:** Burkhard Tümmler.

**Resources:** Lutz Wiehlmann, Claus Urbanke, Sonja Horatzek, Leo Eberl, Burkhard Tümmler.

**Software:** Claus Urbanke.

**Supervision:** Lutz Wiehlmann, Claus Urbanke, Burkhard Tümmler.

**Validation:** Lutz Wiehlmann.

**Visualization:** Lutz Wiehlmann.

**Writing – original draft:** Lutz Wiehlmann, Burkhard Tümmler.

**Writing – review & editing:** Lutz Wiehlmann, Claus Urbanke, Leo Eberl, Burkhard Tümmler.

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
