## [Decision Letter · Decision Letter 0]

3 Apr 2025

PPATHOGENS-D-25-00006

Versatile roles of rubredoxin reductase of Pseudomonas aeruginosa TBCF10839 in virulence and stress protection

PLOS Pathogens

Dear Dr. Tümmler,

Thank you for submitting your manuscript to PLOS Pathogens, which was evaluated by members of the editorial board and three external referees. We apologize for the delay in the review of your manuscript. We feel that your study has merit but does not fully meet PLOS Pathogens's publication criteria as it currently stands. Therefore, we invite you to submit a revised version of the manuscript that addresses the points raised during the review process.

We ask you to focus on Reviewer 1’s comments regarding quantification of survival of the rubB mutant (ideally, of a deletion mutant) and a complemented strain in the PMN phagocytosis assay, on H2O2 dose response curves, and on edits to figure labels and clarifications in figure legends. We also recommend that you describe the limitations of the transcriptome analyses using gene chips with PAO1 as reference, as suggested by Reviewer 2.

Please submit your revised manuscript within 60 days Jun 02 2025 11:59PM. If you will need more time than this to complete your revisions, please reply to this message or contact the journal office at plospathogens@plos.org. Please include the following items when submitting your revised manuscript:

We look forward to receiving your revised manuscript.

Kind regards,

Gregory Priebe

Academic Editor

PLOS Pathogens

D. Scott Samuels

Section Editor

PLOS Pathogens Sumita Bhaduri-McIntosh

Editor-in-Chief

PLOS Pathogens

orcid.org/0000-0003-2946-9497 Michael Malim

Editor-in-Chief

PLOS Pathogens

orcid.org/0000-0002-7699-2064

**Journal Requirements:**

1) Please provide an Author Summary. This should appear in your manuscript between the Abstract (if applicable) and the Introduction, and should be 150-200 words long. The aim should be to make your findings accessible to a wide audience that includes both scientists and non-scientists. Sample summaries can be found on our website under Submission Guidelines:

https://journals.plos.org/plospathogens/s/submission-guidelines#loc-parts-of-a-submission

2) Some material included in your submission may be copyrighted. According to PLOSu2019s copyright policy, authors who use figures or other material (e.g., graphics, clipart, maps) from another author or copyright holder must demonstrate or obtain permission to publish this material under the Creative Commons Attribution 4.0 International (CC BY 4.0) License used by PLOS journals. Please closely review the details of PLOSu2019s copyright requirements here: PLOS Licenses and Copyright. If you need to request permissions from a copyright holder, you may use PLOS's Copyright Content Permission form.

Potential Copyright Issues:

- Please confirm (a) that you are the photographer of Figures 1, 3, and 4, or (b) provide written permission from the photographer to publish the photo(s) under our CC BY 4.0 license.

**Reviewers' Comments:**

Reviewer's Responses to Questions

**Part I - Summary**

Reviewer #1: Wiehlmann and colleagues describe the underappreciated role of rubredoxin reductase (RR) in the virulence of Pseudomonas aeruginosa. Using the CF isolate P. aeruginosa strain TBCF10839, they describe the isolation of a transposon-insertion mutant (rubB::Tn5) with increased susceptibility to neutrophil action. The characterization of this mutant reveals the downregulation of genes and compounds implicated in quorum sensing. They further report increased susceptibility to hydrogen peroxide. The lack of RR also results in decreased virulence in a C. elegans slow-kill assay. The manuscript is well-written and the implication of a role for rubredoxin reductase in P. aeruginosa virulence that goes beyond its canonical function in alkane degradation is intriguing.

However, the data as presented in the manuscript are promising starting points for further investigation but do not always sufficiently back the claims. In some instances, data are not shown. A lack of quantitative rigor for some data sets makes their interpretation difficult. The number of biological and technical replicates should be indicated for all figures.

Reviewer #2: This is an interesting study by Wiehlmann and colleagues describing the how rubredoxin reductase has roles beyond its putative function in P. aeruginosa virulence. It opens of several potential lines of investigation and explains one element of how stress responses and regulatory circuits are interrelated in this bacterium.

Reviewer #3: The manuscript by Wiehlman et al. deals with the identification of the rubredoxin system in virulence and stress response in Pseudomonas aeruginosa. This is a unique contribution to the field. The study is based on the creation of a genome-wide mutant library and screening for reduced virulence in an in vitro system. The screening identified the rub system as responsible for reduced virulence, which was subsequently restored by trans-complementation.

A curious feature is that the rub system is critical for alkane assimilation via oxidation of alkanes to the corresponding alkanol. Thus, finding that the rub system is involved in redox homeostasis, quorum sensing (QS), and virulence is a very unexpected result.

**Part II – Major Issues: Key Experiments Required for Acceptance**

Reviewer #1: The transposon mutant that has previously been isolated in the neutrophil screen should ideally have been recreated as a clean deletion, given that it is central to almost every experiment. Complementing this mutant is helpful, but data for the complement are not always shown.

The rubB transposon mutant was identified previously (Wiehlmann et al., JMM 2007). However, in the 2007 publication, I could not find any details on the severity of the survival defect for this mutant in the PMN phagocytosis assay. Given the importance of this mutant for the current study, the authors should provide a quantification of the mutant survival (ideally, of a deletion mutant of rubB) and its complementation (currently, no information is provided regarding whether complementation rescues the transposon mutant in the survival assay.

Figure 1A: Are these spot colonies? Please provide a scale bar. Ideally, this should be presented as a spot dilution assay or liquid culture growth for a more quantitative assessment.

Figure 1B: Please label the figure. By eye, it is difficult to make a judgment about the coloration of the pellet, especially given the differences in pigmentation of the surrounding culture. Absorption spectra would help compare the strains.

Figure 2A: It took me a while to understand what I was looking at. Please label the P. putida strain on the plate.

Figure 2B and C. It may be sufficient to show Figure 2C.

In figure 2C, the label for the complementation strains obscures where 3-oxo C12-HSL would show up.

Figure 2E, F. The legend indicates that the complementation strain TBCF10839 rubB::Tn5; pME6010::TBrubB is shown. But it is not part of the figure.

Figure 3A. To allow the reader to assess the effect of H2O2 on the three depicted strains, a dose-response curve should be shown for all three strains. This is described in lines 205-211 but the data are not shown.

Line 711-714: The authors write “Within the first 30 seconds P. aeruginosa TBCF10839 (left) reduces hydrogen peroxide to water, whereas TBCF10839 rubB::Tn5 (right) decomposes hydrogen peroxide by disproportionation to produce water and oxygen foam.” Can the authors be certain that TBCF10839 (left) reduces hydrogen peroxide to water in the first 30 seconds? An alternative explanation is that O2 formation is delayed for another reason. Consumption of H2O2 should be monitored to substantiate this claim (e.g. using an assay kit). Please also quantify the data in the supplemental information (bubble formation). How many replicates were performed?

Figure 6: The authors demonstrated the effect of the mutation on QS, which is likely to explain the fast-kill phenotype in C. elegans. Whether this is related to the proposed activity of the rubredoxin reductase in H2O2 protection is not addressed.

Reviewer #2: There is one significant issue:

1. The transcriptomic analysis is strange. The authors determine gene expression in TBCF10839 (and the rubB transposon mutant) to PAO1 using a gene chip. This approach excludes genes specific to TBCF10839 and potentially those genes that do not have 100% identity between the two strains. Also, chip technology has not been the standard for determination of transcriptomes for over a decade.

I don't believe that the transcriptome needs to be re-done but there are several caveats that must be applied to the current approach, which should be included in the text.

Reviewer #3: None

**Part III – Minor Issues: Editorial and Data Presentation Modifications**

Reviewer #1: Please use consistent nomenclature in the figures and text for the mutant and complementation strains.

Reviewer #2: 1. Line 145 - the table of genes that were upregulated might be moved to a supplement so the reader can focus on the down regulated genes.

2. Line 152 - las is not an operon

3. Figures 2BC are difficult to interpret.

4. Figure 6. These data are probably best presented in a K-M survival plot, and should be appropriately statistically analysed.

An additional note: I appreciated the video in the supplement!

Reviewer #3: Suggestions for improvement:

• The Introduction focuses heavily on the alk system and alkane oxidation, but other aspects of Pseudomonas biology are not well addressed. I suggest the authors begin the Introduction by highlighting commonalities between environmental isolates of Pseudomonas aeruginosa and clinical strains. This could include shared carbon source utilization pathways and virulence factors. Then, the authors can describe the rub system as previously reported in the context of alkane degradation.

• In general, avoid the use of abbreviations to start paragraphs or subsection headings.

• Line 129: The comparison between 67 vs. 59 genes is unclear—please clarify.

• Table 1 is organized by PA gene number. It may be more useful to group the genes by function as described in the text. This would help make the long table more digestible.

• Since rubredoxin may interact with multiple proteins, some docking assays could be considered (not necessarily required for this manuscript).

• The potential of rubredoxin as a target for anti-Pseudomonas therapies could be an interesting point to explore in the discussion.

• Line 190: “Transfected” should be replaced with “transformed.”

• Was the whole genome of the rub mutant sequenced?

• Define in the text the selectable marker carried by plasmid pME6010.

PLOS authors have the option to publish the peer review history of their article (what does this mean? ). If published, this will include your full peer review and any attached files.

**Do you want your identity to be public for this peer review?** For information about this choice, including consent withdrawal, please see our Privacy Policy .

Reviewer #1: No

Reviewer #2: No

Reviewer #3: No

**Figure resubmission:**
---

## [Decision Letter · Decision Letter 1]

14 Jul 2025

PPATHOGENS-D-25-00006R1

Versatile roles of rubredoxin reductase of Pseudomonas aeruginosa TBCF10839 in virulence and stress protection

PLOS Pathogens

Dear Dr. Tümmler,

Thank you for submitting your revised manuscript to PLOS Pathogens. We appreciate the thoughtful response to the reviews, but there remain several concerns that you must address before your manuscript is accepted. We invite you to submit another revised version of the manuscript that addresses the points raised during the review process. In addition to a point-by-point response to comments of Reviewer 1, focusing on the rigor of quantification and statistical analyses, we ask you to elaborate on the curve fit methodology and to provide statistical analysis of colony size on hydrogen peroxide (Fig. 4); you will likely need to consult with a statistician regarding the appropriate test to use (which will likely require curve-fitting, regression and root mean square deviation; a two-way ANOVA is not suitable).

Please submit your revised manuscript within 30 days Sep 12 2025 11:59PM. If you will need more time than this to complete your revisions, please reply to this message or contact the journal office at plospathogens@plos.org. Please include the following items when submitting your revised manuscript:

We look forward to receiving your revised manuscript.

Kind regards,

Gregory P Priebe, M.D.

Academic Editor

PLOS Pathogens

D. Scott Samuels

Section Editor

PLOS Pathogens

Sumita Bhaduri-McIntosh

Editor-in-Chief

PLOS Pathogens

orcid.org/0000-0003-2946-9497

Michael Malim

Editor-in-Chief

PLOS Pathogens

orcid.org/0000-0002-7699-2064

**Reviewers' Comments:**

Reviewer's Responses to Questions

**Part I - Summary**

Reviewer #1: line 115: The authors state that cultures reached early stationary phase after 20 h of growth in LB. This seems slow, given that under typical batch-culture conditions (with shaking) P. aeruginosa PAO1 and PA14 usually reach early stationary phase after 5 h of growth. It would be helpful to see growth curves. The timing of growth phases can sometimes be inaccurately estimated if growth curves are not plotted on a logarithmic scale.

Figure 1B, C: Please label x- and y-axes for the RGB plots. They may be better visible on a white background.

Figure 2E and F:

how many data points are averaged in the respective graphs?

please connect data points with straight lines

It is still not clear to me why a difference between the two strains is only observed at ~24-28 hours.

Figure 3A:

Instead of showing a single image, it would be preferable if the author showed growth curves integrating multiple data sets.

Figure 4:

How was the trendline generated? Please indicate the average colony sizes with dedicated symbols and connect them with straight lines.

What do the indicated symbols represent? I assume they are the averages of three technical replicates?

line 263: The authors conclude that the rubredoxin system rescues the bacteria from H2O2 exposure. There may indeed be a modest effect. However, given the spread of data points its difficult to infer if it is statistically significant. Please include a statistical analysis.

Reviewer #2: The authors have addressed my previous concerns adequately

Reviewer #3: The authors have done a very nice work and have dealt satisfactorily with my comments. Thank You

**Part II – Major Issues: Key Experiments Required for Acceptance**

Reviewer #1: (No Response)

Reviewer #2: none

Reviewer #3: The authors have done a very nice work and have dealt satisfactorily with my comments. Thank You

**Part III – Minor Issues: Editorial and Data Presentation Modifications**

Reviewer #1: (No Response)

Reviewer #2: none

Reviewer #3: The authors have done a very nice work and have dealt satisfactorily with my comments. Thank You

PLOS authors have the option to publish the peer review history of their article (what does this mean? ). If published, this will include your full peer review and any attached files.

**Do you want your identity to be public for this peer review?** For information about this choice, including consent withdrawal, please see our Privacy Policy .

Reviewer #1: No

Reviewer #2: No

Reviewer #3: No

**Figure resubmission:**
---

## [Editor Report · Decision Letter 2]

16 Aug 2025

Dear Prof. Tümmler,

We are pleased to inform you that your manuscript 'Versatile roles of rubredoxin reductase of Pseudomonas aeruginosa TBCF10839 in virulence and stress protection' has been provisionally accepted for publication in PLOS Pathogens.

Best regards,

Gregory Priebe

Academic Editor

PLOS Pathogens

D. Scott Samuels

Section Editor

PLOS Pathogens

Sumita Bhaduri-McIntosh

Editor-in-Chief

PLOS Pathogens

orcid.org/0000-0003-2946-9497

Michael Malim

Editor-in-Chief

PLOS Pathogens

orcid.org/0000-0002-7699-2064
---

## [Editor Report · Acceptance letter]

Dear Prof. Tümmler,

We are delighted to inform you that your manuscript, " 

Versatile roles of rubredoxin reductase of Pseudomonas aeruginosa TBCF10839 in virulence and stress protection," has been formally accepted for publication in PLOS Pathogens.

Best regards,

Sumita Bhaduri-McIntosh

Editor-in-Chief

PLOS Pathogens

orcid.org/0000-0003-2946-9497

Michael Malim

Editor-in-Chief

PLOS Pathogens

orcid.org/0000-0002-7699-2064